# Conformal Calibration of Statistical Confidence Sets

**Luben M. C. Cabezas**                                            *lucruz45.cab@gmail.com*
*Department of Statistics and Institute of Mathematics and Computer Science*
*Federal University of São Carlos and University of São Paulo*
*São Carlos, SP 13565-905 and 13566-590, Brazil*
*Université Grenoble Alpes, Inria, CNRS, Grenoble INP, LJK*
*Grenoble, France*

**Guilherme P. Soares**[*]                                       *guilherme.soares.25@ucl.ac.uk*
*Department of Statistical Science*
*University College London*
*London, WC1E 6BT, UK*

**Thiago R. Ramos**                                                   *thiagorr@ufscar.br*
*Department of Statistics*
*Federal University of São Carlos*
*São Carlos, SP 13565-905, Brazil*

**Rafael B. Stern**                                                   *rbstern@gmail.com*
*Institute of Mathematics and Statistics*
*University of São Paulo*
*São Paulo, SP 05508-090, Brazil*

**Rafael Izbicki**                                               *rafaelizbicki@gmail.com*
*Department of Statistics*
*Federal University of São Carlos*
*São Carlos, SP 13565-905, Brazil*

**Reviewed on OpenReview:** *https://openreview.net/forum?id=J4lK62PVE6*

## Abstract

Constructing valid confidence sets is a crucial task in statistical inference, yet traditional methods often face challenges when dealing with complex models or limited observed sample sizes. These challenges are frequently encountered in modern applications, such as Likelihood-Free Inference (LFI). In these settings, confidence sets may fail to maintain a confidence level close to the nominal value. In this paper, we introduce two novel methods, TRUST and TRUST++, for calibrating confidence sets to achieve distribution-free conditional coverage. These methods rely entirely on simulated data from the statistical model to perform calibration. Leveraging insights from conformal prediction techniques adapted to the statistical inference context, our methods ensure both finite-sample local coverage and asymptotic conditional coverage as the number of simulations increases, even if the observed (real) sample size $n$ is small. They effectively handle nuisance parameters and provide computationally efficient uncertainty quantification for the estimated confidence sets. This allows users to assess whether additional simulations are necessary for robust inference. Through theoretical analysis and experiments on models with tractable and intractable likelihoods, we demonstrate that our methods outperform existing approaches, particularly in small-sample regimes. This work bridges the gap between conformal prediction and statistical inference, offering practical tools for constructing valid confidence sets in complex models.

---

[*]Work done while a student at the Institute of Mathematics and Computer Science, University of São Paulo.

# 1 Introduction

Confidence sets are fundamental tools in statistical inference, allowing researchers to constrain the value of a parameter $\theta \in \Theta$ based on observed data $\mathbf{x} \in \mathcal{X}$. A confidence set $R(\mathbf{x}) \subset \Theta$ is considered valid from a frequentist perspective if it satisfies the condition

$$\mathbb{P}\left(\theta \in R(\mathbf{X})|\theta\right) = 1 - \alpha \quad \forall \theta \in \Theta, \tag{1}$$

where $\alpha \in (0, 1)$ is a predefined significance level. This means that $R$ must achieve the correct coverage regardless of the true value of $\theta$.

Traditional methods for constructing confidence sets, such as those detailed in standard textbooks (DeGroot & Schervish, 2012; Schervish, 2012; Casella & Berger, 2024), are often inadequate when applied to complex modern models. For instance, constructing confidence sets for mixture models is notably challenging because standard asymptotic results may not hold (Chen & Li, 2009; Wichitchan et al., 2019). Additionally, many traditional methods rely on asymptotic distributions, making them unsuitable for problems with small sample sizes ($n$).

The challenges become more difficult in Likelihood-Free Inference (LFI) scenarios, where the statistical model is implicitly defined by a complex simulator of $\mathbf{X}|\theta$ and the likelihood function is intractable (Izbicki et al., 2014; Lueckmann et al., 2019; Izbicki et al., 2019; Papamakarios et al., 2019; Cranmer et al., 2020). In such cases, test statistics must be estimated, and as a result, traditional methods for constructing confidence sets often perform poorly (Dalmasso et al., 2024).

This work aims to calibrate confidence sets to achieve conditional coverage (Equation 1), even for challenging models. We focus on confidence sets defined as:

$$R(\mathbf{X}) := \left\{\theta \in \Theta \mid \tau(\mathbf{X}, \theta) \geq C_\theta\right\}, \tag{2}$$

where the cutoff $C_\theta$ is chosen to ensure conditional coverage, and $\tau$ measures the plausibility that $\mathbf{X}$ was generated from $\theta$ (in the language of hypothesis tests, $\tau$ is a test statistic). We assume $\tau$ is fixed (see Sections 1.1.1 and 1.1.2 for details on its selection) and provide tools for calibrating $C_\theta$.

Our calibration process only assumes the availability of a simulated dataset of independent pairs, $\{(\theta_1, \mathbf{X}_1), \ldots, (\theta_B, \mathbf{X}_B)\}$, where each $\theta_b$ is drawn from some reference distribution $r(\theta)$ and each $\mathbf{X}_b$ is generated from the statistical model with parameters $\theta_b$, i.e., $\mathbf{X}_b \sim \mathbf{X} \mid \theta_b$. We assume that $\tau(\mathbf{X}, \theta)|\theta$ is strictly continuous for every $\theta$. Note that each $\mathbf{X}_b$ represents the entire dataset that may be observed, rather than a single sample point.

**Novelty.** While several methods have been proposed to calibrate $C_\theta$ for general problems (Section 1.1), our method offers several new contributions:

1. **Distribution-free guarantees with $B$-asymptotic conditional coverage.** Our approach constructs confidence sets $\widehat{R}_B(\mathbf{X}) := \{\theta \in \Theta \mid \tau(\mathbf{X}, \theta) \geq \widehat{C}_\theta\}$ with finite-sample local coverage:

$$\mathbb{P}\left(\theta \in \widehat{R}_B(\mathbf{X})|\theta \in A\right) = 1 - \alpha,$$

where $A$ is a subset of $\Theta$ chosen so that

$$\mathbb{P}\left(\theta \in \widehat{R}_B(\mathbf{X})|\theta \in A\right) \approx \mathbb{P}\left(\theta \in \widehat{R}_B(\mathbf{X})|\theta\right).$$

Additionally, our method provides $B$-asymptotic conditional coverage:

$$\lim_{B \longrightarrow \infty} \mathbb{P}\left(\theta \in \widehat{R}_B(\mathbf{X})|\theta\right) = 1 - \alpha.$$

Thus, our method is robust to poor estimates of $C_\theta$, offering distribution-free guarantees while maintaining $B$-asymptotic validity. Notably, this coverage is *not* asymptotic with respect to the size

of the observed dataset $\mathbf{x}$, $n$, as is common with traditional asymptotic approximations. Instead, it relies solely on the number of simulations, $B$, which can generally be increased given sufficient computational resources. Furthermore, the set $A$ is chosen to ensure that local coverage approximates conditional coverage.

2. **Uncertainty quantification.** In most cases, except for rare instances where $C_\theta$ can be computed directly (Section 1.1), current methods provide only an estimate $\widehat{C}_\theta$ of the true $C_\theta$. As a result, the confidence intervals produced are only approximations of the true intervals which inherently contain errors. However, these methods do not offer a way to quantify the uncertainty surrounding these estimated intervals. In contrast, our method explicitly constructs confidence sets that provide reliable uncertainty quantification around the estimated confidence intervals. Furthermore, this uncertainty quantification is computationally efficient and does not rely on costly procedures such as bootstrapping. Moreover, it allows users to assess whether additional simulations are necessary for robust inference.

3. **Effective handling of nuisance parameters.** Most models involve nuisance parameters, which are not of direct interest but can complicate inference. Current methods often struggle with properly handling nuisance parameters, particularly when they are numerous, as they typically require expensive numerical maximization algorithms in high-dimensional spaces to ensure coverage (Dalmasso et al., 2024). In contrast, our method significantly reduces the computational burden by transforming the high-dimensional maximization problem into one of finding the maximum over a small, finite set of points, allowing for much more efficient optimization.

Figure 1 compares `TRUST++` with other methods for constructing confidence intervals for the slope coefficient in the generalized linear model explored in Section 5.3. Despite the presence of 3 nuisance parameters, `TRUST++` produces intervals that are closer to the oracle confidence set, both in terms of size and empirical coverage. It even outperforms traditional GLM confidence sets based on $\chi^2$ approximations, as well as Monte Carlo-based intervals.

Our tools are developed using insights from conformal prediction techniques (see Section 1.1 for related work). However, while conformal methods focus on generating prediction sets for new labels $Y$, which are random variables, statistical inference aims to create confidence sets for a fixed parameter $\theta$ that generated the data $\mathbf{x}$. Thus, conformal techniques need to be adapted to fit this purpose. In particular, asymptotic conditional coverage, which is essential for valid sets, is not guaranteed by standard conformal methods.

We provide two methods:

- `TRUST`, which uses regression trees to offer a fast approximation to $C_\theta$.

- `TRUST++`, which enhances `TRUST` by employing bagging of trees to deliver more accurate estimates. The extension from trees to forests is non-trivial due to the need for finite sample guarantees.

The remaining of this work is organized as follows: Section 1.1 contains a discussion on related work. Section 2 details the methodology, including the proposed methods and estimation procedures. Section 3 presents the theoretical results. Section 4 discusses implementation details. Section 5 provides applications to examples with tractable likelihood functions, likelihood-free inference, and nuisance parameter problems. Section 6 offers final remarks. Additional materials are provided in Appendices A through C, including experimental details, supplementary algorithms, and complete theoretical proofs. Finally, all code and supplementary materials are available in the GitHub repository: https://github.com/Monoxido45/CSI

## 1.1 Relation to Other Work

### 1.1.1 Classical Confidence Sets

Confidence sets typically follow the form described in Equation 2. The function $\tau(\mathbf{X}, \theta)$ can be a pivotal quantity, which is a quantity whose distribution does not depend on $\theta$ (e.g., $\sigma^{-1}(X - \mu)$ when $X \sim N(\mu, \sigma)$

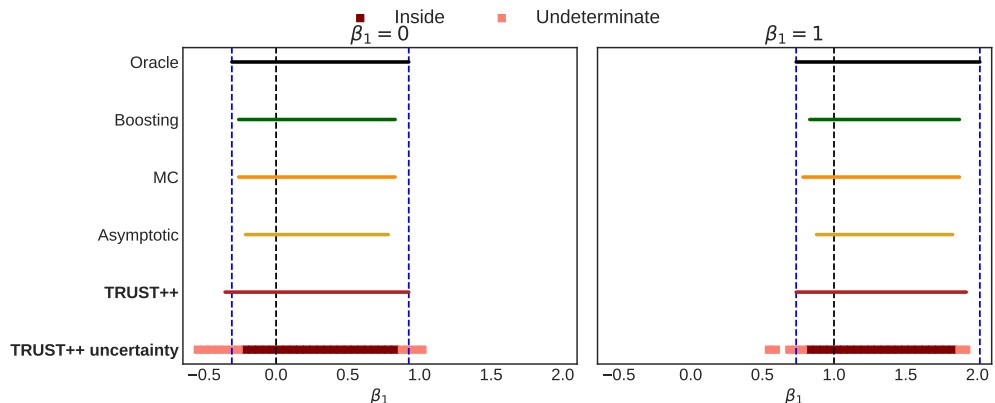

(a) 95% confidence intervals. Blue dotted lines represent the lower and upper bounds for the oracle confidence interval.

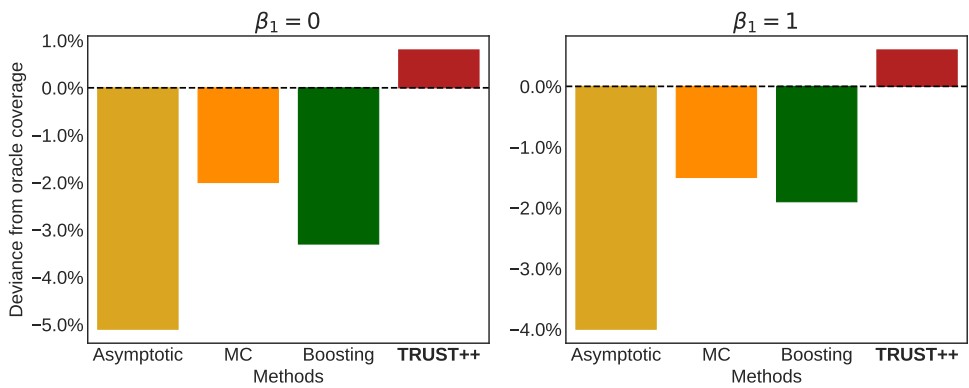

(b) Deviance from oracle coverage.

Figure 1: Confidence intervals and deviance from oracle coverage comparisons for the GLM example with a sample size of $n = 50$. Our method closely approximates the oracle in terms of both the estimated confidence intervals (top) and coverage probabilities (bottom), while also providing uncertainty quantification for the intervals. This feature highlights regions where the true parameter value is more likely to lie. If the undetermined region is large, users can increase the number of simulations to reduce its size and refine the results.

and $\sigma$ is known). Another example of $\tau(\mathbf{X}, \theta)$ is test statistics used for testing the null hypothesis that the data was generated from $\theta$, such as the likelihood ratio statistic $f(\mathbf{x}|\theta)/\sup_{\theta' \in \Theta} f(\mathbf{x}|\theta')$ (Lehmann & Romano, 2005).

A key challenge is computing $C_\theta$. In simple cases, $C_\theta$ can be determined in closed form. For instance, when $\tau$ is a pivotal quantity, $C_\theta$ does not depend on $\theta$ and is typically straightforward to find (Casella & Berger, 2024). More often, asymptotic derivations are employed, which are effective only for large sample sizes $n$ and for certain test statistics (Van der Vaart, 2000). However, these approximations require regularity conditions that frequently fail (Algeri et al., 2019). Similarly, bootstrap methods can be used, but these are effective only for large sample sizes $n$ and also require regularity conditions (Van der Vaart, 2000).

Our methods instead provide consistent (as $B$ increases) estimates of $C_\theta$ for any fixed $\tau$, even for small sample sizes $n$. We only require a simulated dataset $\{(\theta_1, \mathbf{X}_1), \ldots, (\theta_B, \mathbf{X}_B)\}$.

### 1.1.2 Likelihood-Free Inference

Likelihood-Free Inference (LFI) problems are problems in which the statistical model lack a tractable analytical distribution for $\mathbf{X}|\theta$, but it is still possible to generate simulations from it. For a comprehensive review, see Cranmer et al. (2020). In such situations, $\tau$ needs to be defined from the simulated set as well. For instance, one may use $\tau(\mathbf{x}, \theta_0)$ as $\widehat{\mathbb{V}}^{-1}[\theta|\mathbf{x}](\widehat{\mathbb{E}}[\theta|\mathbf{x}] - \theta_0)^2$, where the conditional mean and variance are estimated using the simulated dataset (Masserano et al., 2023). For other possibilities, see e.g. Brehmer et al. (2020); Dalmasso et al. (2024); Carzon et al. (2025b).

In LFI, once $\tau$ has been defined, $C_\theta$ is often estimated via Monte Carlo simulations for each fixed $\theta$, but this approach becomes prohibitively expensive in higher-dimensional spaces. Alternatively, Dalmasso et al. (2020; 2024); Masserano et al. (2023) observed that $C_\theta$ is the $1 - \alpha$ conditional quantile of $\tau(\mathbf{X}, \theta)$ on $\theta$, and thus used quantile regressors to estimate $C_\theta$ using the simulated dataset $\{(\theta_1, \mathbf{X}_1), \ldots, (\theta_B, \mathbf{X}_B)\}$, where $\theta_b$ is drawn from a reference distribution $r(\theta)$ and $\mathbf{X}_b$ is drawn from the statistical model at $\theta_b$. These methods recover $C_\theta$ as $B \to \infty$, but they do not offer finite-sample guarantees. Complementary work has studied diagnostic tools for conditional density models and Bayesian inference algorithms in simulator-based settings (Zhao et al., 2021).

There is also a substantial body of work on Bayesian LFI, but these methods do not achieve conditional in the sense of Eq. 1) even as $B \to \infty$ (Hermans et al., 2022).

### 1.1.3 Conformal Prediction

Conformal methods have recently emerged as an approach to using data to obtain valid prediction regions under minimal assumptions (Vovk et al., 2005; Shafer & Vovk, 2008; Angelopoulos et al., 2023). Specifically, given exchangeable labeled data $\{(\mathbf{X}_1, Y_1), \ldots, (\mathbf{X}_m, Y_m)\}$, these methods create prediction regions $R(\mathbf{x})$ such that

$$\mathbb{P}\left(Y_{m+1} \in R(\mathbf{X}_{m+1})\right) \geq 1 - \alpha. \tag{3}$$

The first step in a standard conformal method involves creating a conformity score $\hat{s} : \mathcal{X} \times \mathcal{Y} \to \mathbb{R}$, which measures how well the outputs of a regression model predict new labels $Y \in \mathcal{Y}$. In a regression setting, a standard choice for the conformity score is the regression residual, given by $\hat{s}(\mathbf{x}, y) = |y - \widehat{\mathbb{E}}[Y|\mathbf{x}]|$. The prediction region then takes the form $R(\mathbf{x}) = \{y : \hat{s}(\mathbf{x}, y) \leq t\}$, where $t$ is chosen to be the adjusted $(1 - \alpha)$-quantile of the residuals $\hat{s}(\mathbf{x}, y)$ evaluated on a holdout set not used to train $\hat{s}$.

The function $\tau$ used to construct confidence sets can be interpreted as a conformal score $\hat{s}$, and $\theta$ can be interpreted as $Y$. However, statistical theory is typically employed to design $\tau$ in a way that yields optimal confidence sets, leveraging the known distribution $f(\mathbf{x} \mid \theta)$. Therefore, standard conformal scores are suboptimal for designing $\hat{s}$ for confidence sets. Additionally, most conformal scores only deal with univariate $Y$, whereas $\theta$ is generally multivariate (see however Dheur et al. (2024) for approaches that can be used for multivariate $Y$'s).

Another key difference between prediction intervals and confidence sets is that $\theta$ is not random, whereas $Y$ is. Thus, marginal coverage (Eq. 3) is insufficient for statistical inference problems; typically, coverage for all fixed $\theta$'s, corresponding to conditional coverage (Eq. 1), is also desired. This is because the distribution $r(\theta)$, used to sample $\theta$ on the training set, is often arbitrary; it does not encode any randomness observed in the real world. Unfortunately, it is well-known that conditional coverage can't be achieved without strong assumptions (Lei & Wasserman, 2014). While many conformal approaches aim to achieve conditional coverage asymptotically, they only control coverage conditional on the features $\mathbf{x}$, and not $y$. The exception to this are label-conditional conformal prediction methods (Vovk et al., 2014; 2016; Sadinle et al., 2019; Ding et al., 2023), but these apply only to classification problems.

Finally, we note that conformal sets have recently been applied to statistical inference (Patel et al., 2024; Baragatti et al., 2026; Cabezas et al., 2025a). While these methods are highly useful, they are designed for a Bayesian framework, where the focus is on controlling marginal coverage over the prior distribution.

### 1.1.4 Partition-Based Conformal Methods

Our approach to asymptotically controlling conditional coverage is closely related to the works of Vovk et al. (2005); Lei & Wasserman (2014); Boström & Johansson (2020); Izbicki et al. (2020; 2022); Vovk et al. (2022); Santos et al. (2026). These methods first partition the feature space and then calibrate $t$ using the conformal approach described in the previous section for each partition element separately. In particular, TRUST is closely related to LoCart (Cabezas et al., 2025b), which creates a data-driven partition of $\mathcal{X}$ using regression trees designed to ensure that

$$\mathbb{P}\left(Y_{n+1} \in R(\mathbf{X}_{n+1}) \mid \mathbf{X}_{n+1}\right) \approx 1 - \alpha.$$

However, instead of partitioning the feature space $\mathcal{X}$, we partition the parameter space $\Theta$ to achieve coverage conditions on $\theta$.

### 1.1.5 Conformal Predictive Distributions

Conformal Predictive Distributions (CPDs) are a recent extension of the conformal prediction framework (Vovk et al., 2019; 2020; Jonkers et al., 2024). These aim to model the entire predictive distribution of $Y$ in a non-parametric way. If a conformal $\hat{s}$ is isotonic and balanced (Vovk et al., 2020), the CPD for a new test instance $\mathbf{X}_{n+1}$ is defined as:

$$Q((\mathbf{X}_{n+1}, y), u) := \frac{1}{n} \sum_{i=1}^{n} \mathbb{I}(\hat{s}(\mathbf{X}_i, Y_i) \leq \hat{s}(\mathbf{X}_{n+1}, y)) + \frac{u}{n}, \tag{4}$$

for each $y \in \mathbb{R}$ and where $u \sim U(0, 1)$ serves as a correction term to ensure the continuity of $Q((X_{n+1}, y), u)$.

In simple terms, $Q$ provides an estimated probability distribution for the test labels $y$ based on conformity scores. Specifically, if the data is i.i.d., $Q$ is valid, meaning that $Q((\mathbf{X}_{n+1}, y), u)$ follows a uniform distribution (Vovk et al., 2019), and thus $Q$ is a valid cumulative distribution function. Leveraging the validity of $Q$, we construct $1 - \alpha$ confidence CP prediction intervals by utilizing the specific $\alpha/2$ and $1 - \alpha/2$ percentiles of the CPD (Boström et al., 2021).

In this work, we use Conformal Predictive Distributions to estimate the distribution of the test statistic $\tau(\mathbf{X}, \theta)$. This approach provides the desired distribution-free properties when estimating $C_\theta$. Additionally, our CPDs allow for the computation of p-values for hypotheses of interest. However, directly using the definition from Eq. 4 for statistical inference problems will not lead to consistent intervals for any fixed $\theta$. The percentiles of $Q$ would not depend on $\theta$, which is the same issue discussed in Section 1.1.3. To address this, we employ a local CPD based on a partition of $\Theta$. We note that Boström et al. (2021) also constructs CPDs based on partitions, but these partitions are made in $\mathcal{X}$ rather than in $\Theta$. While this approach is valuable for prediction tasks, it does not address the problem of statistical inference.

## 2 Methodology

Fix a test statistic $\tau$, and let $H(\cdot|\theta)$ denote the distribution of $\tau(\mathbf{X}, \theta)$ given $\theta$, that is, for each $\tau_0 \in \mathbb{R}$, let

$$H(\tau_0|\theta) := \mathbb{P}\left(\tau(\mathbf{X}, \theta) \leq \tau_0|\theta\right).$$

Now, notice that for the confidence set from Equation 2 to have a $(1 - \alpha)$ confidence level, $C_\theta$ must satisfy

$$1 - \alpha = \mathbb{P}(\theta \in R(\mathbf{X})|\theta) = \mathbb{P}(\tau(\mathbf{X}, \theta) \geq C_\theta|\theta).$$

In other words, we need $\mathbb{P}(\tau(\mathbf{X}, \theta) \leq C_\theta) = \alpha$, which implies that $C_\theta$ is connected to the CDF $H$ by

$$C_\theta = H^{-1}(\alpha|\theta),$$

where $H^{-1}(\cdot|\theta)$ denotes the generalized inverse of $H(\cdot|\theta)$, given by

$$H^{-1}(\lambda) = \inf\{t \in \mathbb{R} : H(t|\theta) \geq \lambda\}.$$

This suggests we derive confidence sets by estimating $H$ and then using the plugin cutoff

$$\widehat{C}_{\theta,B} = \widehat{H}_B^{-1}(\alpha|\theta),$$

where $\widehat{H}_B$ is the adjusted empirical distribution function defined as

$$\widehat{H}_B(t|\theta) := \frac{1}{B+1}\left(\sum_{b=1}^{B}\mathbf{1}\{\tau(\mathbf{X}^{(b)},\theta)\leq t\}+1\right).$$

The estimated confidence set will then be

$$\widehat{R}_B(\mathbf{X}) := \left\{\theta\in\Theta \mid \tau(\mathbf{X},\theta)\geq\widehat{C}_{\theta.B}\right\}.$$

The estimated conditional CDF $\widehat{H}_B$ can also be used to test statistical hypothesis of the type $H_0 : \theta = \theta_0$. Specifically, p-values can be estimated via

$$\widehat{H}_B(\tau(\mathbf{x}_{\mathrm{obs}},\theta_0)|\theta_0),$$

where $\mathbf{x}_{\mathrm{obs}}$ represents the observed data[1].

We build on this by introducing a novel approach to estimate $H(\cdot|\theta)$, which ensures robust properties.

## 2.1 Estimating $H(\cdot|\theta)$

Our first estimator of $H(\cdot|\theta)$, TRUST (Tree-based Regression for Universal Statistical Testing), is constructed by partitioning $\Theta$. First, we show how any partition of $\Theta$ can be used to form an estimator of $H$. Following that, we discuss how to optimally select such a partition.

### 2.1.1 Partition-based Estimate

Let $\mathcal{A}$ be any fixed partition of $\Theta$ and $\{(\theta_1,\mathbf{X}_1),\ldots,(\theta_B,\mathbf{X}_B)\}$ be the simulated dataset from the statistical model. The cumulative distribution function $H(\cdot|\theta)$ can be estimated using the empirical cumulative distribution of the $\theta_b$ values from the simulated set that fall into the same partition element as $\theta$. Formally, for each $t\in\mathbb{R}$, we define

$$\widehat{H}_B(t|\theta,\mathcal{A}) := \frac{1}{|I_{A(\theta)}|+1}\left(\sum_{b\in I_{A(\theta)}}\mathbb{I}\left(\tau(\mathbf{X}_b,\theta_b)\leq t\right)+1\right), \tag{5}$$

where $A(\theta)$ represents the element of $\mathcal{A}$ containing $\theta$, and $I_{A(\theta)} = \{b\in\{1,\ldots,B\} : \theta_b\in A(\theta)\}$ is the set of indices for all $\theta_b$ values that fall into $A(\theta)$.

With this partition-based construction of $\widehat{H}$, the plugin cutoff $\widehat{C}_{\theta,B} = \widehat{H}_B^{-1}(\alpha|\theta)$, used to estimate confidence sets, corresponds to the adjusted $\alpha$-quantile of the values $\{\tau(\mathbf{X}_b,\theta_b) : b\in I_{A(\theta)}\}$. Also, $\widehat{H}_B$ is essentially a conditional predictive conformal distribution (Section 1.1.5), with the exception that we do not add the uniform distribution used to ensure continuity as it is not needed here. This method therefore aligns with a partition-based conformal approach. However, unlike standard partition-based conformal methods, the partition $\mathcal{A}$ must be selected to approximate $H(\alpha|\theta)$ well. The next section discusses how to choose $\mathcal{A}$ to achieve this goal.

### 2.1.2 Choosing the partition: TRUST

For each $\theta$, our goal is to estimate the cutoff $C_\theta = H^{-1}(\alpha \mid \theta)$ so that $\mathbb{P}\{\tau(X,\theta)\leq C_\theta \mid \theta\}\approx\alpha$. With a finite simulation budget, estimating this quantile separately for every $\theta$ is noisy, so TRUST *pools* simulations across

---

[1]Indeed, under the null hypothesis, $\mathbb{P}\left(H(\tau(\mathbf{X}_{\mathrm{obs}},\theta_0))|\theta_0)\leq\alpha|\theta_0\right) = \mathbb{P}\left(\tau(\mathbf{X}_{\mathrm{obs}},\theta_0)\leq H^{-1}(\alpha|\theta_0)|\theta_0\right) = H\left(H^{-1}(\alpha)\right) = \alpha$, and therefore $\widehat{H}_B(\tau(\mathbf{x}_{\mathrm{obs}},\theta_0)|\theta_0)$ will be a valid p-value if $H$ is consistently estimated as $B\longrightarrow\infty$.

parameter values that induce similar distributions of $\tau(X, \theta)$. Concretely, `TRUST` learns a data-adaptive partition $\mathcal{A}$ of $\Theta$ and, for a target $\theta$, computes $\widehat{C}_{\theta,B}$ as the adjusted empirical $\alpha$-quantile of the $\tau$-values whose $\theta_b$ fall in the same cell $A(\theta) \in \mathcal{A}$. The partition must balance (i) having enough points per cell to stabilize quantile estimation and (ii) keeping cells homogeneous enough that pooling does not bias the cutoff.

To choose $\mathcal{A}$, `TRUST` fits a regression tree to the simulated pairs $(\theta_b, \tau(\mathbf{X}_b, \theta_b))_{b=1}^B$ and uses the induced leaves as the partition elements. Intuitively, the tree creates cells $A$ within which the conditional law of $\tau(\mathbf{X}, \theta)$ varies slowly with $\theta$, so that the pooled empirical CDF in (5) provides an accurate proxy for $H(\cdot \mid \theta)$. With such a partition, the guaranteed local coverage $\mathbb{P}\{\theta \in \widehat{R}_B(\mathbf{X}) \mid \theta \in A(\theta)\} = 1 - \alpha$ serves as a good approximation to the desired conditional coverage $\mathbb{P}\{\theta \in \widehat{R}_B(\mathbf{X}) \mid \theta\} \approx 1 - \alpha$ (Meinshausen & Ridgeway, 2006, Theorem 1).

Although our theoretical results assume that the data used to construct the optimal partition is separate from the data used to build the empirical CDF in Equation 5 (refer to Section 3 for details), in practice, we do not perform this data split, as it did not yield any performance gains in our experiments. Algorithm 1 outlines the complete `TRUST` procedure, and Figure 2 provides a graphical visualization of it.

---

**Algorithm 1:** `TRUST` algorithm

**Data:** Simulated dataset $(\mathbf{X}_1, \theta_1), \dots, (\mathbf{X}_B, \theta_B)$; a grid $\Theta_{grid} \subseteq \Theta$

**Result:** The confidence set $\widehat{R}_B(\mathbf{X})$

compute $\mathcal{I} = (\theta_1, \tau(\mathbf{X}_1, \theta_1)), \dots, (\theta_B, \tau(\mathbf{X}_B, \theta_B))$;

split $\mathcal{I}$ in $\mathcal{I}_{train}$ and $\mathcal{I}_{cal}$, where $\mathcal{I}_{train} \cap \mathcal{I}_{cal} = \emptyset$;

fit a decision tree $\widehat{g}$ with $\mathcal{I}_{train}$: $\widehat{g}(\theta) \approx \mathbb{E}[\tau(\mathbf{X}, \theta) \mid \theta]$;

$\widehat{g}$ will create a partition $\widehat{\mathcal{A}}_{\text{train}} = \{A_1, \dots, A_K\}$ of $\Theta$;

**for** $\theta \in \Theta_{grid}$ **do**

    $A(\theta) \leftarrow$ element of $\widehat{\mathcal{A}}_{\text{train}}$ where $\theta$ falls;

    $J_{A(\theta)} \leftarrow \{i \in [\mathcal{I}_{cal}] : \theta_i \in A(\theta)\}$;

    $\widehat{C}_{\theta,B} \leftarrow$ adjusted $\alpha$-quantile of $\{\tau(X_c, \theta_c)\}_{c \in J_{A(\theta)}}$;

**end**

**return** $\widehat{R}_B(\mathbf{X}) = \left\{\theta^* \in \Theta_{grid} \mid \tau(\mathbf{X}, \theta^*) \geq \widehat{C}_{\theta^*,B}\right\}$

---

## 2.2 From trees to forests: `TRUST++`

A single regression tree can yield partitions that are sensitive to small perturbations of the simulated data. To stabilize the partitioning of $\Theta$, `TRUST++` uses an ensemble of trees to define a data-adaptive notion of similarity between parameter values. However, using standard random-forest machinery to estimate conditional quantiles—for example by averaging tree-wise leaf distributions or by computing tree-wise cutoffs $\widehat{C}_{\theta,B}^{(k)}$ and averaging them across trees—does not, in general, preserve the distribution-free guarantees of partition-based conformal constructions: once the cutoff is averaged across trees, it is no longer an adjusted order statistic computed from an exchangeable set within a fixed cell, and the usual partition-based validity argument no longer applies (Cabezas et al., 2025b).

Instead, `TRUST++` uses the forest only to define a data-adaptive notion of locality in $\Theta$. We first fit a random forest with $K$ regression trees, each trained on a bootstrap sample of the simulated pairs $\{(\theta_b, \tau(\mathbf{X}_b, \theta_b))\}_{b=1}^B$. Let $\rho(\theta, \theta')$ denote Breiman's proximity measure (Breiman, 2001), defined as the number of trees for which $\theta$ and $\theta'$ fall in the same terminal node. Large $\rho(\theta, \theta')$ indicates that many independently grown trees repeatedly group $\theta$ and $\theta'$ together, suggesting that $\tau(\mathbf{X}, \theta)$ has a similar conditional distribution at these parameter values.

**Strict partition (distribution-free version).** We first define a partition $\mathcal{A}$ induced by the equivalence relation $\theta \sim \theta' \iff \rho(\theta, \theta') = K$, i.e., $\theta$ and $\theta'$ belong to the same cell iff they share a leaf in every tree. We then estimate $H(\cdot \mid \theta)$ by the partition-based empirical CDF in (5) and set $\widehat{C}_{\theta,B} = \widehat{H}_B^{-1}(\alpha \mid \theta)$.

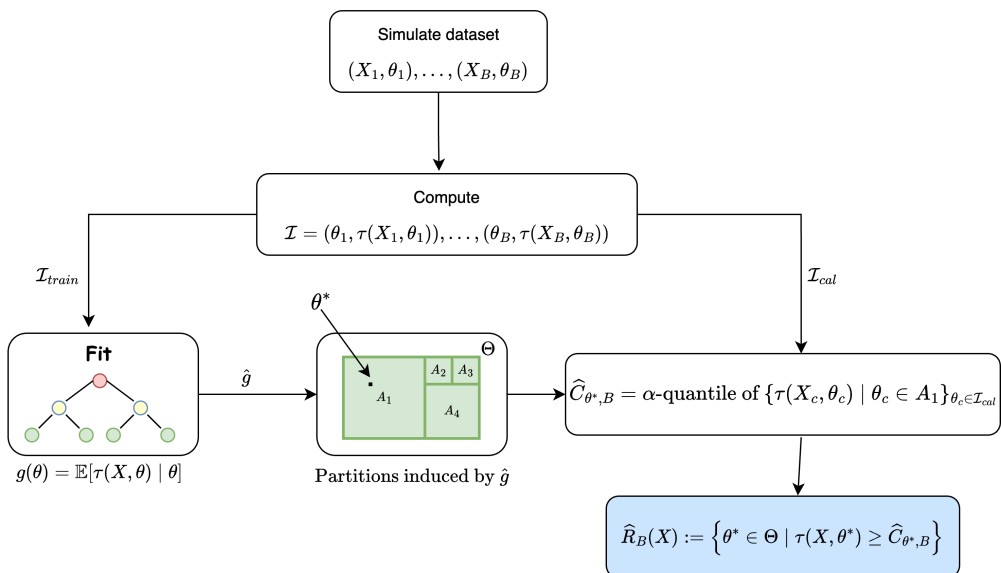

Figure 2: Schematic flowchart of TRUST. Over a simulated dataset $(\mathbf{X}_1, \theta_1), \ldots, (\mathbf{X}_B, \theta_B)$, we compute the statistic $\tau$ and create the set $\mathcal{I}$, which is splitted between $\mathcal{I}_{train}$ and $\mathcal{I}_{cal}$. $\mathcal{I}_{train}$ is used to train a regression tree that predicts $\tau(\mathbf{X}, \theta)$ using $\theta$. The tree partitions the parametric space in $A_1, \ldots, A_k$. If a $\theta^*$, from the grid, falls inside the $A_1$ partition then, in order to calculate the cutoff $\widehat{C}_{\theta^*, B}$, we will compute the adjusted $\alpha$-quantile of all $\tau(\mathbf{X}, \theta)$'s from which $\theta$ belongs both to $\mathcal{I}_{cal}$ and $A_1$. Finally, we use the calculated cutoff to find the confidence set $\widehat{R}_B$.

**Relaxed neighborhoods (practical version).** In practice, we often use the neighborhood $A(\theta) := \{\theta' \in \Theta : \rho(\theta, \theta') \geq M\}$, where $M \in \{1, \ldots, K\}$ controls how aggressively we pool simulations across $\theta$'s. For $M < K$, $A(\theta)$ need not induce a partition and exact finite-sample coverage is not guaranteed (Guan, 2023), but we find it performs well empirically; see Section 4 for how we tune $M$.

TRUST++ can also be framed as a specific instance of the LCP framework by utilizing a box kernel with the Breiman proximity measure replacing the commonly used Euclidean distance. This adaptation allows us to leverage the conformalization procedures from Guan (2023) and Hore & Barber (2023) to transform TRUST++ into a proper conformal method, thereby ensuring exact coverage control. In this setting, the tuning parameter $M$ serves as a kernel bandwidth, regulating the granularity of the partition $\mathcal{A}$. Larger $M$ yields smaller, stricter neighborhoods (less pooling, lower bias, higher variance), while smaller $M$ yields larger neighborhoods (more pooling, higher bias, lower variance).

Algorithm 2 summarizes the non-conformalized version of our method, which we name TRUST++, and Figure 3 shows the flowchart of the procedure.

## 2.3 Nuisance parameters

Many statistical models contain parameters that affect the data distribution but are not of direct interest. Write the full parameter as $\theta = (\mu, \nu)$, where $\mu \in M$ is the parameter of interest and $\nu \in N$ is a nuisance parameter. Our goal is to report a confidence set for $\mu$ only, while maintaining validity uniformly over the nuisance parameter.

**Oracle cutoff for $\mu$.** Fix a test statistic $\tau(\mathbf{X}, \mu)$ (larger values indicate higher plausibility of $\mu$). For each $(\mu, \nu)$, let

$$H(t \mid \mu, \nu) := \mathbb{P}(\tau(\mathbf{X}, \mu) \leq t \mid \mu, \nu), \qquad C_{\mu, \nu} := H^{-1}(\alpha \mid \mu, \nu).$$

If we were interested in inference for the full parameter $(\mu, \nu)$, we would use the cutoff $C_{\mu, \nu}$. However, to guarantee coverage for *all* nuisance configurations while reporting uncertainty only for $\mu$, we must protect

---

**Algorithm 2:** `TRUST++` algorithm

---

**Data:** Simulated dataset $(\mathbf{X}_1, \theta_1), \ldots, (\mathbf{X}_B, \theta_B)$; a grid $\Theta_{grid} \subseteq \Theta$

**Result:** The confidence set $\widehat{R}_B(\mathbf{X})$

compute $\mathcal{I} = (\theta_1, \tau(\mathbf{X}_1, \theta_1)), \ldots, (\theta_B, \tau(\mathbf{X}_B, \theta_B))$;

split $\mathcal{I}$ in $\mathcal{I}_{train}$ and $\mathcal{I}_{cal}$, where $\mathcal{I}_{train} \cap \mathcal{I}_{cal} = \emptyset$;

draw $K$ bootstrapped samples from $\mathcal{I}_{train}$;

fit $K$ `TRUST` trees, each one with a different sample (Alg. 1);

**for** $\theta \in \Theta_{grid}$ **do**

    $A(\theta) \leftarrow \{\theta' \in \Theta \mid \rho(\theta', \theta) \geq M\}$, where $\rho(\theta', \theta)$ is the Breiman's proximity measure and M a tuning parameter;

    $J_{A(\theta)} \leftarrow \{i \in [\mathcal{I}_{cal}] : \theta_i \in A(\theta)\}$;

    $\widehat{C}_{\theta,B} \leftarrow$ adjusted $\alpha$-quantile of $\{\tau(\mathbf{X}_c, \theta_c)\}_{c \in J_{A(\theta)}}$;

**end**

**return** $\widehat{R}_B(\mathbf{X}) = \left\{ \theta^* \in \Theta_{grid} \mid \tau(\mathbf{X}, \theta^*) \geq \widehat{C}_{\theta^*,B} \right\}$

---

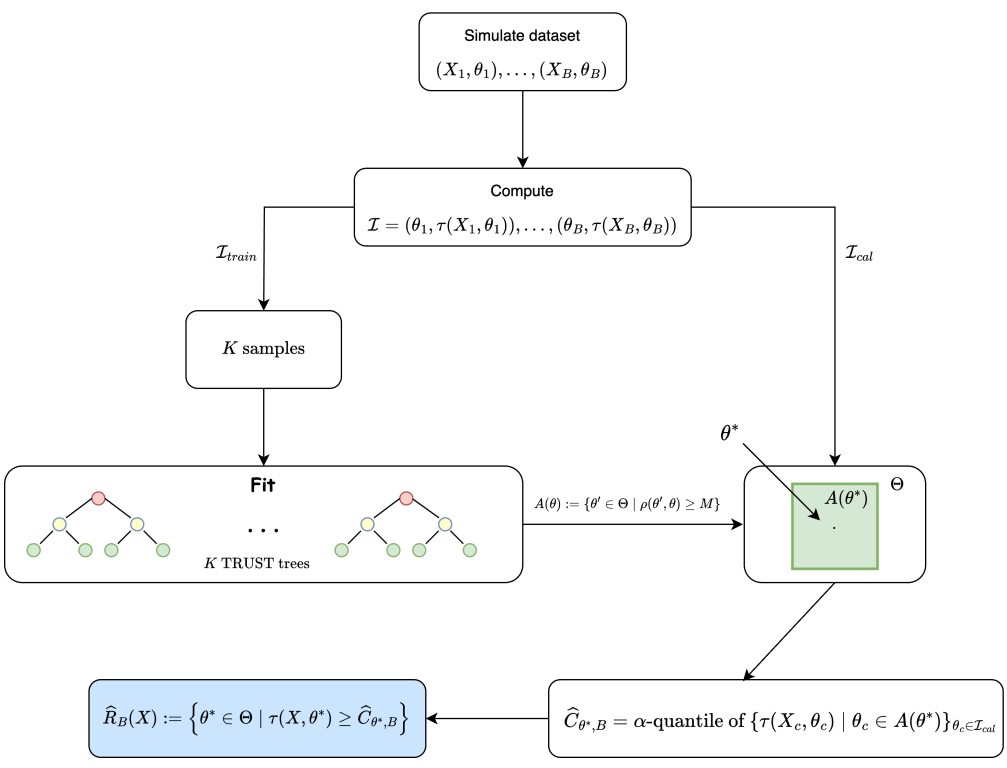

Figure 3: Schematic flowchart of `TRUST++`. Over a simulated dataset $(\mathbf{X}_1, \theta_1), \ldots, (\mathbf{X}_B, \theta_B)$, we compute the statistic $\tau$ and create the set $\mathcal{I}$. Then, we draw K bootstrapped samples over the split $\mathcal{I}_{train}$ and use each sample to fit a different `TRUST` tree. The set $A(\theta)$ is calculated using Breiman's proximity measure, $\rho$, which defines whether two $\theta, \theta'$ belong to the same partition element. The cutoff $\widehat{C}_{\theta^*,B}$ is then calculated as the $\alpha-$quantile of all $\tau$ statistics of their respective $\theta$ that falls over $A(\theta)$, for the $\theta$'s that belong to $\mathcal{I}_{cal}$. Lastly, the estimated confidence set $\widehat{R}_B$ can be calculated using $\widehat{C}_{\theta^*,B}$.

against the worst-case $\nu$, leading to the oracle cutoff (Dalmasso et al., 2024; Izbicki, 2025)

$$C_\mu = \inf_{\nu \in N} C_{\mu,\nu}. \tag{6}$$

The resulting confidence set is

$$R(\mathbf{x}) := \{\mu \in M : \tau(\mathbf{x}, \mu) \geq C_\mu\}.$$

**Estimating $C_\mu$ with TRUST / TRUST++.** We estimate $C_{\mu,\nu}$ using the same machinery as before, but now treating $(\mu, \nu)$ as the covariate. Given simulations $\{((\mu_b, \nu_b), \mathbf{X}_b)\}_{b=1}^B$, we fit a tree/forest regressing $\tau(\mathbf{X}_b, \mu_b)$ on $(\mu_b, \nu_b)$, which induces a partition $\mathcal{A}$ of $\Theta = M \times N$ into cells (leaves).

For any leaf $A \in \mathcal{A}$, define the leaf-wise empirical CDF (as in (5)) and its adjusted $\alpha$-quantile; denote this leaf cutoff by $q_A$. Then, for a given $(\mu, \nu)$, our estimator satisfies

$$\widehat{C}_{\mu,\nu,B} = q_{A(\mu,\nu)},$$

i.e., it depends only on the leaf $A(\mu, \nu)$ containing $(\mu, \nu)$. Finally, we estimate the nuisance-robust cutoff by

$$\widehat{C}_{\mu,B} = \inf_{\nu \in N} \widehat{C}_{\mu,\nu,B}. \tag{7}$$

**Why the infimum becomes a finite minimum.** Because $\widehat{C}_{\mu,\nu,B}$ is *constant within each leaf*, the map $\nu \mapsto \widehat{C}_{\mu,\nu,B}$ is piecewise constant when $\mu$ is fixed. Hence, the infimum in (7) reduces to a minimum over the finitely many leaves intersecting the slice $\{\mu\} \times N$:

$$\widehat{C}_{\mu,B} = \min\Big\{q_A : A \in \mathcal{A}, \ A \cap (\{\mu\} \times N) \neq \emptyset\Big\}.$$

This replaces a potentially expensive continuous optimization over $\nu$ by a discrete minimization over a small set of candidate leaves.

**Example 1** (One-dimensional nuisance parameter)**.** *Assume $\nu$ is one-dimensional. For a fixed $\mu$, the tree induces finitely many intervals in $\nu$ (e.g., $\nu < s_1$, $s_1 \leq \nu < s_2$, $\nu \geq s_2$), each corresponding to a leaf with a constant cutoff $q_1, q_2, q_3$. Then*

$$\inf_\nu \widehat{C}_{\mu,\nu,B} = \min\{q_1, q_2, q_3\},$$

*so computing $\widehat{C}_{\mu,B}$ only requires evaluating one cutoff per relevant leaf.*

**Efficient candidates for TRUST++.** For TRUST++, the same principle applies: $\widehat{C}_{\mu,\nu,B}$ changes only when $(\mu, \nu)$ crosses a split threshold in one of the trees. A practical strategy is therefore to build a small candidate set of nuisance values from the split thresholds. Let $\mathbb{A} \subset N$ be the set of nuisance-coordinate split values appearing in the fitted forest (optionally restricted to splits near the root). Construct a candidate grid

$$\mathbb{G} := \{\nu \pm \epsilon : \nu \in \mathbb{A}\},$$

where $\epsilon$ is chosen small enough so that $\nu - \epsilon$ and $\nu + \epsilon$ fall on opposite sides of each split (e.g., one third of the minimum coordinate-wise separation among distinct split values). Then, for each fixed $\mu$, we approximate (7) by

$$\widehat{C}_{\mu,B} \approx \min_{\nu \in \mathbb{G}} \widehat{C}_{\mu,\nu,B}.$$

This avoids gridding $N$ directly (which becomes infeasible in moderate/high dimensions) and leverages the fact that only split-induced regions matter for the leaf-wise cutoffs.

## 2.4 Uncertainty About the Confidence Sets

Our confidence set $\widehat{R}_B(\mathbf{X}) := \{\theta \in \Theta \mid \tau(\mathbf{X}, \theta) \geq \widehat{C}_\theta\}$ is an estimate of the oracle set $R(\mathbf{X}) := \{\theta \in \Theta \mid \tau(\mathbf{X}, \theta) \geq C_\theta\}$, as is the estimate from the other approaches described in Section 1.1. In this section, we explore another advantage of TRUST and TRUST++: They offer a computationally efficient way to approximate the uncertainty associated with these sets, which we describe in what follows.

Recall that, for each $\theta$, $C_\theta$ is estimated by computing the adjusted $\alpha$-quantile of the set $T_{A(\theta)} = \{\tau(\mathbf{X}_b, \theta_b) : b \in I_{A(\theta)}\}$, while $C_\theta$ itself is the $\alpha$-quantile of the distribution of $\tau(\mathbf{X}, \theta)|\theta$. Let $\tau_{A(\theta)}^{(i)}$ be the $i$-th order

statistics of $T_{A(\theta)}$ and define $Z := |\{\delta \in T_{A(\theta)} : \delta \leq C_\theta\}|$, where $Z$ represents the number of statistics in $T_{A(\theta)}$ that are less than or equal to the cutoff $C_\theta$. Note that if the statistics in $T_{A(\theta)}$ are identically distributed, then $Z \sim \text{Binomial}(n, 1 - \beta)$. Using this result, we adapt the method by Hahn & Meeker (2011) for constructing confidence intervals for quantiles to create a $1 - \beta$ confidence set for $C_\theta$. This confidence set is defined as:

$$[\tau_{A(\theta)}^{(l)}, \tau_{A(\theta)}^{(u)}], \tag{8}$$

where $l$ and $u$ are selected to satisfy constraints ensuring both the validity and informativeness of the confidence interval. Specifically, $u$ and $l$ are selected such that $l \leq u \leq |I_{A(\theta)}|$ are as close as possible and satisfy the restriction

$$\mathbb{P}(l \leq Z \leq u - 1) \geq 1 - \beta, \tag{9}$$

yielding a valid confidence interval. This is because

$$\mathbb{P}(\tau_{A(\theta)}^{(l)} \leq C_\theta \leq \tau_{A(\theta)}^{(u)}) = \mathbb{P}(l \leq Z \leq u - 1) \geq 1 - \beta.$$

Thus, by choosing $u$ and $l$ as narrowly as possible satisfying the restriction from Eq. 9 we get the thinnest valid confidence interval for $C_\theta$. Additionally, despite the i.i.d assumption about statistics in $T_{A(\theta)}$ being not strictly true, the values in $T_{A(\theta)}$ are approximately identically distributed due to the way the partition was chosen (Section 2.1.2).

For each $\theta \in \Theta$, let $(\widehat{C}_\theta^L, \widehat{C}_\theta^U)$ represent the resulting confidence interval for $C_\theta$. The uncertainty can be propagated to the estimated sets by calculating the following three sets:

$$\mathcal{I}(\mathbf{X}) = \{\theta \in \Theta \mid \tau(\mathbf{X}, \theta) \geq \widehat{C}_\theta^U\},$$
$$\mathcal{O}(\mathbf{X}) = \{\theta \in \Theta \mid \tau(\mathbf{X}, \theta) \leq \widehat{C}_\theta^L\}, \text{ and}$$
$$\mathcal{U}(\mathbf{X}) = \{\theta \in \Theta \mid \widehat{C}_\theta^L \leq \tau(\mathbf{X}, \theta) \leq \widehat{C}_\theta^U\},$$

where $\mathcal{I}(\mathbf{X})$ represents the set of parameter values confidently inside the confidence interval, $\mathcal{O}(\mathbf{X})$ contains values confidently outside, and $\mathcal{U}(\mathbf{X})$ includes those where the status remains uncertain. In the terminology of 3-way hypothesis testing (Berg, 2004; Esteves et al., 2016; Izbicki et al., 2025), $\mathcal{I}(\mathbf{X})$ is the acceptance region, $\mathcal{O}(\mathbf{X})$ is the rejection region, and $\mathcal{U}(\mathbf{X})$ is the agnostic region. The size of $\mathcal{U}(\mathbf{X})$ can be decreased by increasing the number of simulations used to estimate $C_\theta$, $B$. The next theorem shows that this approach controls the probability of incorrect conclusions.

**Theorem 1.** *Fix $\mathbf{x} \in \mathcal{X}$ and $\theta \in \Theta$. Let $(\widehat{C}_\theta^L, \widehat{C}_\theta^U)$ represent a $(1 - \beta)$-level confidence interval for $C_\theta$ such that $\mathbb{P}\left(C_\theta \leq \widehat{C}_\theta^L\right) = \mathbb{P}\left(\widehat{C}_\theta^U \leq C_\theta\right)$. Then*

$$\mathbb{P}\left(\theta \in \mathcal{I}(\mathbf{x}) | \theta \notin R(\mathbf{x})\right) \leq \beta/2$$

*and*

$$\mathbb{P}\left(\theta \in \mathcal{O}(\mathbf{x}) | \theta \in R(\mathbf{x})\right) \leq \beta/2.$$

This method can be readily extended to the setting with nuisance parameters. Let $\eta^*$ be the configuration of nuisance parameters for which $\widehat{C}_{\mu,B}$ is computed (Equation 7). By construction, $\widehat{C}_{\mu,B}$ is the adjusted $\alpha$-quantile of the set $T_{A(\mu,\eta^*)} = \{\tau(\mathbf{X}_b, \mu_b, \eta_b) : b \in I_{A(\mu,\eta^*)}\}$. Thus, we can derive a confidence set for the optimal cutoff using the same approach as in Eq. 8, based now on the set $T_{A(\mu,\eta^*)}$.

## 3 Theoretical Results

The key aspect of our method lies in the fact that achieving a good approximation of $H(\cdot|\theta)$ allows the partitions in `TRUST` and `TRUST++` to effectively capture the local behavior of the test statistic $\tau$.

Building on this, our theoretical framework relies on ensuring that $\widehat{H}_B$ closely approximates $H$ in both `TRUST` and `TRUST++`. This idea is formalized in the following assumption:

**Assumption 1.** *Let $\widehat{H}_B$ represent the approximation of $H$ under either* TRUST *or* TRUST++. *For any $\varepsilon > 0$ and $\delta > 0$, there exists a $B_0 \in \mathbb{N}$ such that, for all $B \geq B_0$,*

$$\mathbb{P}\left(\sup_{t \in \mathbb{R}, \theta' \in \Theta} \left|\widehat{H}_B(t|\theta') - H(t|\theta')\right| \leq \varepsilon\right) \geq 1 - \delta.$$

This assumption ensures that, with high probability, $\widehat{H}_B$ approximates $H$ as $B$ increases. This result is supported by theoretical findings, such as those of Meinshausen & Ridgeway (2006); Biau et al. (2008), on the consistency of tree-based models. To guarantee this consistency, certain conditions regarding the distribution of covariates and the structure of the trees are necessary. In tree construction, it is important that node sizes are balanced, with the proportion of observations in each node decreasing as the total number of observations grows. Additionally, each variable must have a minimum probability of being selected for node splitting, and splits should ensure a balanced distribution of observations between subnodes. These assumptions are quite reasonable and enable the conditional distribution estimates to be consistent with the true distribution, providing a solid theoretical foundation for our method.

## 3.1 Partition Coverage Guarantees

As discussed in Section 2.1.1, given any fixed partition $\mathcal{A}$ of $\Theta$, the cumulative distribution function $H(\cdot|\theta)$ can be estimated empirically using the values of $\theta_b$ that fall within the same partition element as $\theta$. Although a data-agnostic partitioning approach does not guarantee that $\widehat{H}$ will closely approximate $H$, we can still ensure that the plugin cutoff $\widehat{C}_{\theta,B} = \widehat{H}_B^{-1}(\alpha|\theta)$, corresponding to the $\alpha$-quantile of the values $\{\tau(\mathbf{X}_b, \theta_b) : b \in I_{A(\theta)}\}$, achieves the desired coverage when conditioned on the partition element. This result is formalized in the following theorem.

**Theorem 2** (Partition-Based Coverage). *Let $\{(\theta_1, \mathbf{X}_1), \ldots, (\theta_B, \mathbf{X}_B)\}$ be an i.i.d. simulated dataset with a fixed partition $\mathcal{A}$ of $\Theta$, where each $\theta_b$ is drawn from a reference distribution $r(\theta)$ and each $\mathbf{X}_b$ is generated according to the statistical model with parameters $\theta_b$. For a test statistic $\tau(\mathbf{X}, \theta)$, consider the confidence set constructed by our method:*

$$\widehat{R}_B(\mathbf{X}) = \{\theta \in \Theta \mid \tau(\mathbf{X}, \theta) \geq \widehat{C}_{\theta,B}\},$$

*where $\widehat{C}_{\theta,B} = \widehat{H}_B^{-1}(\alpha|\theta)$ represents the adjusted $\alpha$-quantile of the values $\{\tau(\mathbf{X}_b, \theta_b) : b \in I_{A(\theta)}\}$, conditioned on $\theta$ belonging to the partition element $A(\theta)$. Then, we have*

$$\mathbb{P}(\theta \in \widehat{R}_B(\mathbf{X}) \mid \theta \in A(\theta)) \geq 1 - \alpha,$$

*ensuring the desired coverage probability of the confidence set, conditioned on the partition element.*

This result applies broadly to any partition, yet it carries particular significance for those formed using TRUST and TRUST++. For these methods, we specifically anticipate the relationship

$$1 - \alpha = \mathbb{P}\left(\theta \in \widehat{R}_B(\mathbf{X}) \mid \theta \in A(\theta)\right) \approx \mathbb{P}\left(\theta \in \widehat{R}_B(\mathbf{X}) \mid \theta\right),$$

which suggests that, with these methods, the probability that $\theta$ lies within the estimated region $\widehat{R}_B(\mathbf{X})$, given its association with $A(\theta)$, closely aligns with the nominal confidence level $(1 - \alpha)$. In the following section, we will state the theorem formally. Also, notice that although the local guarantee depends on the choice of $r(\theta)$ (and thus has a Bayesian flavor), the conditional one does not.

Beyond the conditional coverage established above, these results also imply marginal coverage:

**Corollary 1** (Marginal Coverage). *Let $\{(\theta_1, \mathbf{X}_1), \ldots, (\theta_B, \mathbf{X}_B)\}$ be an i.i.d. simulated dataset with a fixed partition $\mathcal{A}$ of $\Theta$, where each $\theta_b$ is drawn from a reference distribution $r(\theta)$ and each $\mathbf{X}_b$ is generated according to the statistical model with parameters $\theta_b$. For a test statistic $\tau(\mathbf{X}, \theta)$, consider the confidence set constructed by our methods. Then $\mathbb{P}(\theta \in \widehat{R}_B(\mathbf{X})) \geq 1 - \alpha$.*

### 3.2 `TRUST` and `TRUST++` Conditional Coverage Guarantees

Both `TRUST` and `TRUST++` employ regression trees that satisfy the consistency Assumption 1, as guaranteed by established results for tree-based models (Meinshausen & Ridgeway, 2006; Biau et al., 2008). With these partitions, we not only guarantee partition-based coverage, as indicated in Theorem 2, but also ensure that both methods asymptotically achieve optimal conditional coverage, as formalized in the next theorem.

**Theorem 3** ($B$-Asymptotic Conditional Coverage). *Let $\{(\theta_1, \mathbf{X}_1), \ldots, (\theta_B, \mathbf{X}_B)\}$ be an i.i.d. simulated dataset, where each $\theta_b$ is drawn from a reference distribution $r(\theta) > 0$ and each $\mathbf{X}_b$ is generated according to the statistical model with parameters $\theta_b$. For a test statistic $\tau(\mathbf{X}, \theta)$, consider the confidence set constructed by either* `TRUST` *or* `TRUST++` *with $M = K$:*

$$\widehat{R}_B(\mathbf{X}) := \{\theta \in \Theta \mid \tau(\mathbf{X}, \theta) \geq \widehat{C}_{\theta, B}\},$$

*where $\widehat{C}_{\theta, B}$ is the cutoff determined by the partition, representing the adjusted $\alpha$-quantile of the distribution of $\tau(\mathbf{X}_b, \theta_b)$ values within the partition containing $\theta$. Then, if Assumption 1 holds, both* `TRUST` *and* `TRUST++` *are asymptotically consistent, that is:*

$$\lim_{B \to \infty} \mathbb{P}\left(\theta \in \widehat{R}_B(\mathbf{X}) \mid \theta\right) = 1 - \alpha.$$

As described in Section 2, in practice we may use relaxed neighborhoods controlled by $M$. When $M < K$, these neighborhoods need not induce a partition of $\Theta$, so exact finite-sample coverage is not guaranteed (Guan, 2023). However, in the asymptotic regime $B \to \infty$, it is natural to consider the partitioning case $M = K$, since the partitions are then expected to be sufficiently populated and hence non-degenerate; this helps explain why our asymptotic analysis is predictive of the behavior observed for the relaxed ($M < K$) version in the experiments.

Theorem 3 shows that the partition constructed by `TRUST` and `TRUST++` is designed so that local coverage closely approximates conditional coverage. Moreover, unlike conventional asymptotic approaches, which typically rely on the sample size of the observed dataset $\mathbf{x}$, $n$, our approach is independent of $n$. Instead, it depends solely on the number of simulations $B$, which can be scaled up given sufficient computational resources. This framework enables both `TRUST` and `TRUST++` to achieve robust, distribution-free guarantees and asymptotically attain optimal conditional coverage as $B$ increases.

In the appendix, we discuss the key results required for the theoretical framework, provide intuition for why our methods work, and present the proofs of the main results.

## 4 Implementation Details and Tuning Parameters

To implement both `TRUST` and `TRUST++`, we use the efficient decision tree and random forest implementations provided by *scikit-learn* (Pedregosa et al., 2011). Since our approaches use regression trees to partition $\Theta$, managing tree growth is crucial to prevent empty or redundant partition elements. We achieve this through a combination of pre- and post-pruning: in both methods, pre-pruning is executed by fixing the `min_samples_leaf` hyperparameter—typically to 100 or 300 samples—to ensure that each leaf is sufficiently well-populated to produce accurate local estimations of $H(\cdot|\theta)$. Specifically, we set 150 and 300 as default samples for `TRUST` and `TRUST++`, respectively, throughout all main experiments (Sections 5.1 and 5.2).

While pre-pruning provides a baseline for leaf stability, `TRUST` further refines its partition through cost-complexity post-pruning, which balances complexity with predictive performance by removing less informative nodes. In contrast, we omit post-pruning in `TRUST++` to preserve the diversity across the ensemble, as excessive pruning can diminish the benefits of aggregating varied partitions. For all other hyperparameters, we adhere to *scikit-learn* defaults, with the exception of the random forest size, which we set to 200 trees to ensure a robust ensemble.

The impact of these structural choices is further explored in Appendix A.6, where our sensitivity analysis (Figure 15) reveals that `min_samples_leaf` is the most consequential hyperparameter for `TRUST++` performance. While this value could scale with sample size, we found that fixing it at a substantial level, such as

300, effectively stabilizes the model. This stability allows the tuning focus to shift toward the neighborhood parameter $M$, yielding the robust results displayed in Section 5. Notably, TRUST avoids these sensitivity issues because its cost-complexity post-pruning step acts as an automated regularizer, naturally restricting hyperparameter selections toward those that optimize predictive performance.

Shifting the focus to this neighborhood selection, our default configuration for the tuning parameter is $M = K/2$, placing TRUST++ in a majority-vote regime. In this setup, $\theta'$ is included in the neighborhood $A(\theta)$ only if a majority of trees assign both points to the same leaf. Although this intuitive choice performs well across the LFI problems discussed in Section 5.2, it is not universally optimal, as some tasks may necessitate larger neighborhoods to compensate for the potential approximate non-invariance of the underlying statistics.

To address this, we propose a straightforward grid-search algorithm for optimizing $M$ using a small, additional simulated validation grid. The main idea is to select $M$ from a fine grid between 0 and $K$ that minimizes estimated deviation from conditional coverage. This is done by calculating coverage across the validation grid with a batch of statistics simulated at each fixed grid point and then computing the mean absolute error of the estimated coverage relative to the nominal confidence level $1 - \alpha$. Section 5 provides further details on this process. Alternatively, coverage can be estimated using the LF2I diagnostic module (Dalmasso et al., 2024), which also leverages an additional simulation set. Algorithm 3 in Appendix B outlines the tuning procedure for $M$.

## 5    Applications

In this section, we compare the coverage performance of TRUST and TRUST++ (both tuned and majority-votes versions) to the other state-of-the-art competing methods on tractable likelihood, likelihood-free, and nuisance parameter settings.

To assess performance, we use several statistic and likelihood/simulator combinations across multiple sample sizes $n$ and simulation budgets $B$ in both likelihood-based and likelihood-free scenarios. In the nuisance parameter setting, we benchmark our methods with two examples: one using likelihood-free inference and another based on likelihood. We set a common confidence level of $1 - \alpha = 0.95$ across all experiments. Comparisons are made against the following baseline methods:

- **Gradient Boosting Quantile Regression (Boosting)**: implemented in the *scikit-learn* library (Pedregosa et al., 2011), we use it to estimate $C_\theta$ through the $(1 - \alpha)$ conditional quantile of the $\tau(\mathbf{X}, \theta)$ given $\theta$ (Dalmasso et al., 2020). We fix the maximum depth as 3, the tolerance for early stopping as 15 and the maximum iteration as 100. The remaining hyperparameters are fixed as *scikit-learn*'s default. To control tree growth in each iteration, we limit the maximum depth to 3 (instead of the default, which is unlimited) and employ a weak learner ensemble approach, which is well-suited for boosting methods (Freund & Schapire, 1997; Hastie et al., 2009).

- **Monte-Carlo (MC)**: implemented with an equally spaced grid over $\Theta$, we simulate $n_{MC}$ statistics for each grid element and estimate $C_\theta$ using the $(1 - \alpha)$-quantile of these simulations at the closest grid point. For multi-dimensional $\Theta$, the grid is formed by combining equally spaced one-dimensional grids obtained along each coordinate of $\Theta$. To ensure comparability with other methods, the uni-dimensional grid size is set to $\left\lceil \frac{B}{n_{MC}}^{1/d} \right\rceil$, where $d = \dim \Theta$. This guarantees that the Monte-Carlo simulation budget in multi-dimensional problems will be close to the fixed budget $B$. We set in all cases $n_{MC} = 500$.

- **Asymptotic**: this approach relies on classic asymptotic results for certain test statistics. It estimates $C_\theta$ as the $(1 - \alpha)$-quantile of the statistic's asymptotic invariant distribution. Since estimated statistics in the LFI setting lack invariant asymptotic approximations, this method is only applied for comparison in some scenarios with tractable likelihoods.

- **Normalizing-flows (nflow)**: we use conditional neural spline flows (Durkan et al., 2019) to estimate the distribution of the test statistic conditional on $\theta$. Once trained, the $(1 - \alpha)$-quantile $C_\theta$ is estimated by drawing a large sample $B_{quantile}$ from this learned distribution and computing

the empirical quantile. Further details regarding the architecture and training procedures for this quantile estimator are provided in Appendix A.

To evaluate the conditional coverage performance of each approach in both likelihood-based and likelihood-free scenarios, we compute the Mean Absolute Error (MAE) of each method's conditional coverage concerning the confidence level $(1 - \alpha)$. We begin by estimating the conditional coverage of each method through the simulation of several statistics inside an evaluation grid. Let $\Theta_{\text{grid}}$ denote such a grid. We compute the coverage for each $\theta' \in \Theta_{\text{grid}}$ at level $\alpha$ for any cutoff estimation method $\widehat{C}$ as follows:

$$\text{cover}_\alpha(\widehat{C}, \theta') := \frac{1}{n_{\text{sim}}} \sum_{i=1}^{n_{\text{sim}}} \mathbb{I}\left( \tau(\theta', \mathbf{X}_i^{(\theta')}) \leq \widehat{C}_{\theta'} \right) , \qquad (10)$$

where $n_{\text{sim}}$ represents the number of simulated statistics and $\mathbf{X}_i^{(\theta')}$ denotes the $i$-th observation simulated under $\theta'$. We then define the MAE of the method's conditional coverage relative to the nominal confidence level $(1 - \alpha)$ as:

$$\text{MAE}(\widehat{C}, \alpha) := \frac{1}{|\Theta_{\text{grid}}|} \sum_{\theta' \in \Theta_{\text{grid}}} \left| \text{cover}_\alpha(\widehat{C}, \theta') - (1 - \alpha) \right| . \qquad (11)$$

The MAE serves as an effective metric for quantifying the extent to which each method $\widehat{C}$ deviates from the nominal level of conditional coverage.

In the nuisance scenarios, we assess performance by measuring each method's coverage deviation from the oracle coverage, defined as follows:

$$d_\alpha(\widehat{C}, C) := \frac{1}{|\Theta_{grid}|} \sum_{\theta' \in \Theta_{grid}} \left| \text{cover}_\alpha(\widehat{C}, \theta') - \text{cover}_\alpha(C, \theta') \right| , \qquad (12)$$

where $C$ is the oracle cutoff specified in Eq. 6. This oracle cutoff controls coverage across all parameters—main, and nuisance —by accounting for worst-case variations in the nuisance parameter. Consequently, we compare each method's coverage against the oracle rather than the nominal level, as the oracle typically over-covers for the parameters of interest to ensure reliable coverage across all nuisance parameters.

We replicate each experiment 50 times for likelihood-based, 30 times for likelihood-free, and 15 times for nuisance examples, and compute the average MAE or deviation from the oracle and its standard error. We identify the top methods in each case as those with the lowest average error, followed by any other methods with performance not significantly different from the best (determined by overlapping 95% asymptotic confidence intervals for the MAE).

In our primary evaluations, the likelihood-based scenarios involve one to two parameters, while the LFI settings extend up to five dimensions, reflecting standard inference benchmarks. To assess the scalability of our framework beyond these common regimes, we provide an extensive analysis of how our methods manage increasing parameter dimensionality in Appendix A.7. Furthermore, detailed MAE and standard error visualizations for the comprehensive set of benchmark experiments are provided in Appendix A.5.

## 5.1 Examples with tractable likelihood function

In this section, we apply our methods to construct confidence sets on three standard statistical models. We always assume the data is $\mathbf{X} = (X_1, \ldots, X_n)$, where the $X_i$'s are i.i.d. To generate the data, we use the Normal model with fixed variance, the Gaussian mixture model (GMM), and the Lognormal model with both mean and scale as parameters. For the choice of $\tau$, we explore the following statistics: Likelihood ratio (Drton, 2009); Kolmogorov-Smirnov (Marsaglia et al., 2003); Frequentist Bayes Factor (Dalmasso et al., 2024; Carzon et al., 2025a) and E-value (Pereira & Stern, 1999). The detailed setup is described in Appendix A.1.1.

To evaluate the comparative performance of each method, we consider sample sizes $n \in \{10, 20, 50, 100\}$ and simulation budgets $B \in \{1000, 5000, 10000, 15000\}$ across all statistic and likelihood combinations. While

Figure 4 provides a global summary of these results, we specifically highlight the performance for the BFF statistic in Figure 5. Detailed comparisons and heatmaps for the remaining statistics across all experimental settings are provided in Appendix A.5.1.

Figures 4, 5, and the supplementary visualizations in Appendix A.5.1 reveal several critical performance distinctions among the evaluated methods:

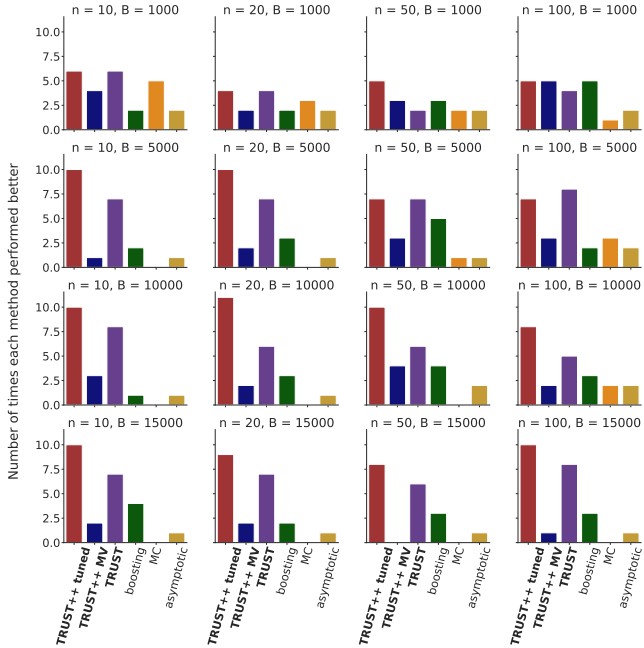

Figure 4: Tractable Problems: Frequency with which each method achieved the best performance (lowest significant MAE) across all statistic–likelihood combinations and varying $n$ and $B$. Our methods (in bold) consistently outperform competitors, with a clear advantage in smaller-sample settings ($n = 10, 20$), where asymptotic approaches tend to underperform. For disaggregated results, see Appendix A.5.1.

- At $B = 1000$, there are no clear advantage among the methods for different sample sizes $n$, with `TRUST` and tuned `TRUST++` showing competitive results in these scenarios. This balance may be attributed to the limited number of simulation samples, which reduces the ability to differentiate among methods regarding cutoff estimation and overall performance.

- Tuned `TRUST++` consistently outperforms all competing methods across $B$ from 5000 to 15000 and for all $n$.

- `TRUST` exhibits good performances for all $n$, showing a higher count of scenarios with superior outcomes across each combination of $n$ and $B$.

- Both Monte-Carlo and Asymptotic methods present comparatively poor performance when the simulation budget exceeds 1000.

- While the boosting method performs well in specific $(n, B)$ configurations, it is generally surpassed by `TRUST` and the tuned `TRUST++`. Although these gains are somewhat modest, they are frequent across the majority of tested settings, suggesting more robust behavior from our tree-based approaches. These findings are corroborated by Figures 4 and 5, as well as the case-by-case analysis in Appendix A.5.1, which show that these incremental improvements persist across a wide range of test statistics.

- Examining the individual statistics (Figure 5 and Appendix A.5.1), our methods show superior performance for the KS statistic, most notably `TRUST++` (tuned) and `TRUST`. For the LR statistic, both the asymptotic approach and our methods achieve satisfactory results. Furthermore, our

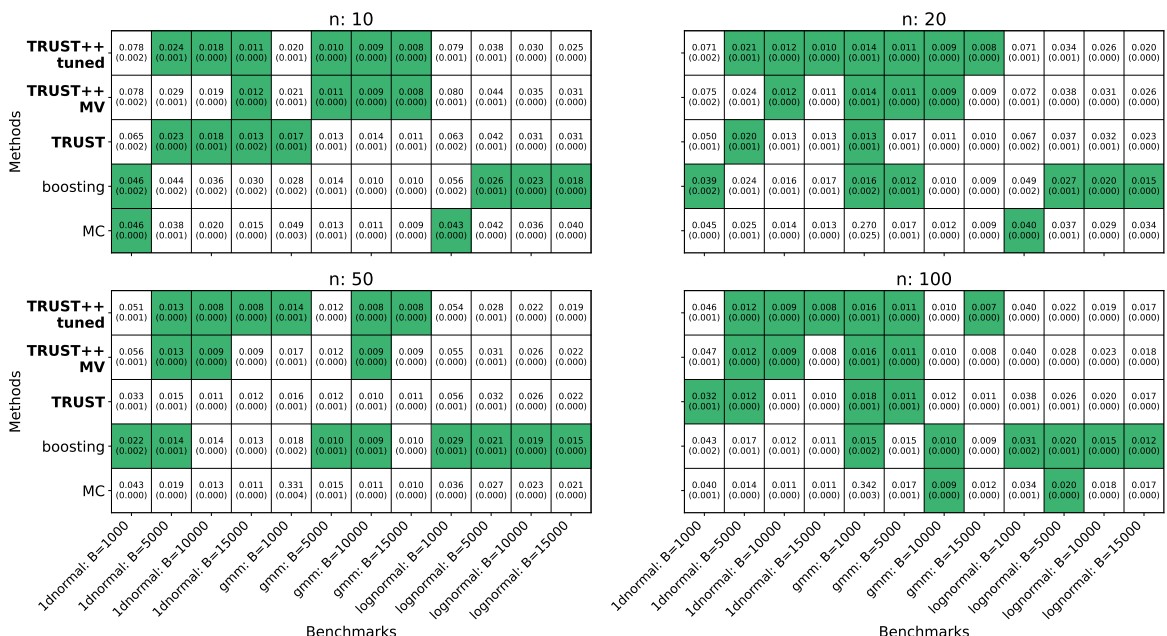

Figure 5: Conditional coverage MAE heatmap for the BFF statistic across tractable benchmarks. Rows denote the evaluated methods, while columns represent specific combinations of data-generating processes and simulation budgets $B$. Tuned `TRUST++`, `TRUST++`-MV, and boosting consistently demonstrate the lowest MAE, representing the most robust choices across the tested scenarios.

> methods exhibit good performance for BFF and E-value, with boosting representing a competitive alternative in these settings, mainly in the lognormal scenario.

Based on all comparisons, we conclude that our framework consistently outperforms competing approaches in terms of conditional coverage for several settings.

## 5.2 Likelihood-free Inference Problems

We now apply our methods to construct confidence sets across five well-studied LFI benchmarks, where the likelihood is intractable or no analytical formula exists for certain statistics, requiring them to be estimated. In cases of intractable likelihood, we simulate the data $\mathbf{X} = (X_1, \ldots, X_n)$ using a high-fidelity simulator $F_\theta$. Given a fixed $\theta$, we also assume that each $X_i$ is i.i.d. The simulators we use are the following: SLCP (Simple likelihood complex posterior), M/G/1, Weinberg (Hermans et al., 2022); Two Moons (Greenberg et al., 2019); SIR (Lueckmann et al., 2021). We consider the following choices of $\tau$: the E-Value, BFF, and Waldo (Masserano et al., 2023). A detailed description of them can be found in the Appendix A.1.2.

Since an exact posterior $f(\theta|\mathbf{x})$ is unavailable in these likelihood-free contexts, we approximate it using a Neural Posterior Estimator (NPE) (Hermans et al., 2022). While we previously introduced Normalizing Flows as a baseline for modeling the distribution of test statistics, we here utilize a distinct flow architecture to directly estimate the parameter posterior. This method approximates $f(\theta|\mathbf{x})$ through an amortized estimator $\hat{f}_\psi(\theta|\mathbf{x})$ built with neural network-based bijective transformations (Rezende & Mohamed, 2015). The estimator is trained on a simulated sample $\{(\theta_1, \mathbf{X}_1), \ldots, (\theta_B, \mathbf{X}_B)\}$ generated from each simulator. By distinguishing this posterior estimation from the statistical flow used for $C_\theta$ calculation, we benchmark the robustness of `TRUST++` against varied applications of neural density estimation. Details on the architecture and simulation budgets are given in Appendix A.

For each statistic and simulator, we evaluate all methods across combinations of $n \in \{1, 5, 10, 20\}$ and $B \in \{10, 15, 20, 30\} \times 10^3$, increasing the simulation budgets relative to Section 5.1 to account for the

greater complexity of these problems. While Figure 6 provides a global summary of these performances, we specifically include a heatmap for the BFF statistic in Figure 7 to illustrate results for this case in the LFI setting. Detailed comparisons of coverage results, disaggregated by the remaining statistics and each data-generating process, are provided in Appendix A.5.2. The results again reveal several performance distinctions among the methods:

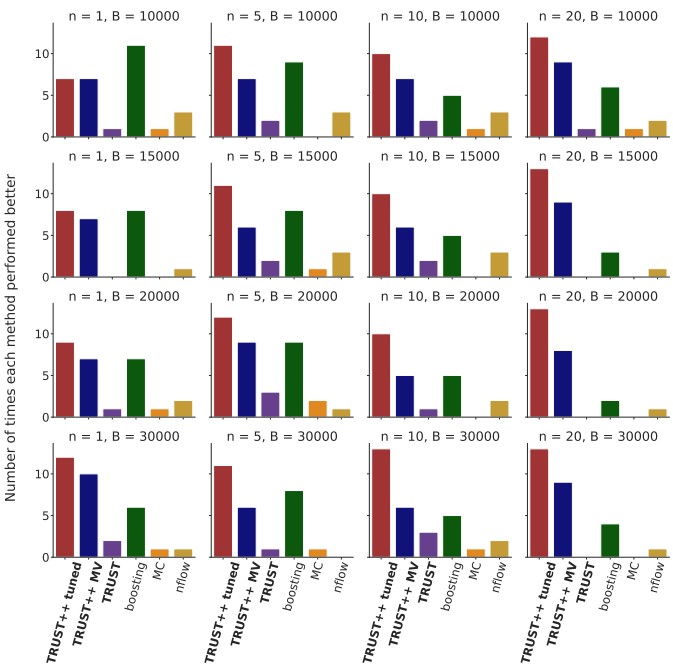

Figure 6: Likelihood-free Inference Problems: Frequency with which each method achieved the best result for each statistic–simulator combination across varying sample sizes $n$ and simulation budgets $B$. Our methods (in bold) show consistent superiority when $B > 10,000$ and remain competitive even at $B = 10,000$. For disaggregated results, see Appendix A.5.2.

- For $n \in \{1, 5\}$ and simulation budgets below $30,000$, Figure 6 indicates a relatively balanced performance between tuned TRUST++, TRUST++ MV, and boosting. However, in all scenarios with $n \in \{10, 20\}$, tuned TRUST++ consistently achieves superior results, while TRUST++ MV outperforms the boosting baseline, albeit by a small margin. This suggests that our Breiman distance approach for computing local cutoffs adapts effectively to likelihood-free settings as sample sizes and simulation budgets increase.

- Figure 6 indicates that tuned TRUST++ outperforms competing methods in nearly all combinations (14 out of 16). Again, tuned TRUST++ exhibits a high count of combinations compared to all other approaches, mirroring its performance observed in tractable experiments.

- The Monte Carlo method performs poorly across nearly all scenarios, showing weaker coverage control than other methods.

- The boosting method demonstrates competitive coverage control and consistent performance in several scenarios, notably outperforming in the specific case of $n = 1$ and $B = 10000$. However, in most cases, it is outperformed by both tuned TRUST++ and TRUST++ MV.

- Despite exhibiting poorer coverage control than boosting and tuned TRUST++, TRUST++ MV still shows good performance across all scenarios, which suggests that this default choice for $M$ is considerably robust on LFI applications.

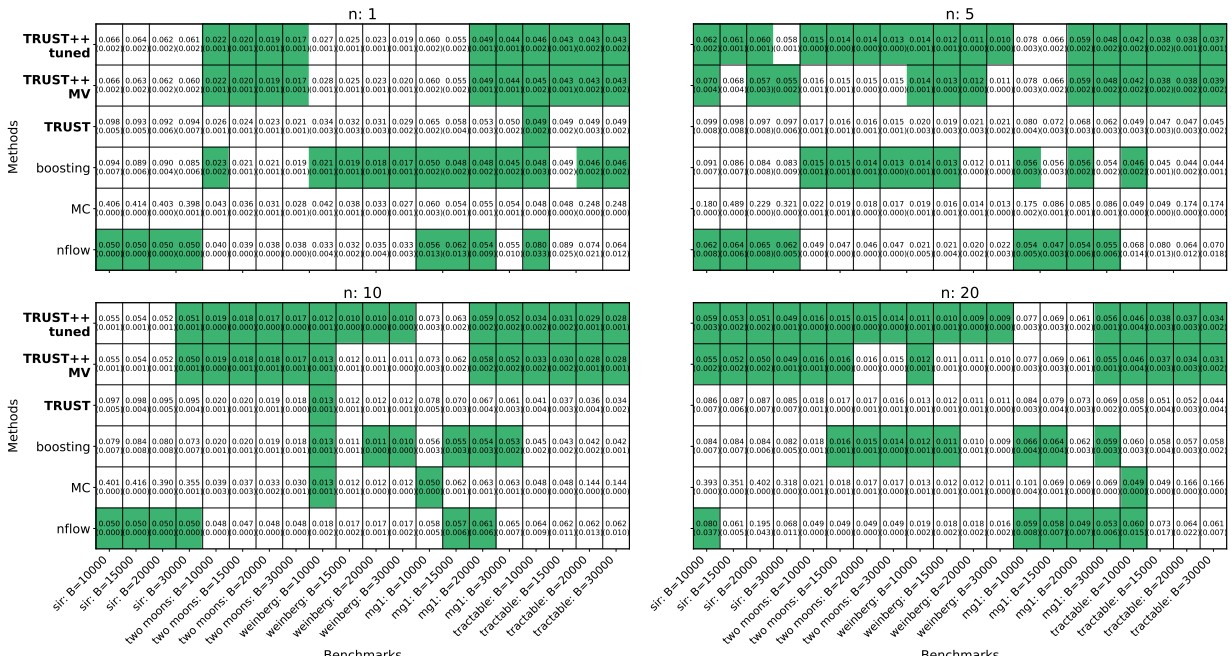

Figure 7: Conditional coverage MAE heatmap for the BFF statistic across LFI benchmarks. Rows and columns represent methods and simulator/budget combinations, respectively. Tuned `TRUST++`, `TRUST++`-MV, and boosting are the strongest competitors. The nflow baseline is specifically competitive for SIR and M/G/1 benchmarks.

- In contrast to the results seen in tractable experiments, `TRUST` demonstrates subpar performance and coverage control compared to both boosting and `TRUST++` in likelihood-free scenarios. This indicates that the `TRUST` partition requires enhancements for likelihood-free contexts, a need effectively addressed through the ensembling approach used in `TRUST++`.

- The Normalizing Flows baseline demonstrates competitive performance in specific settings, such as the SIR and M/G/1 benchmarks at small sample sizes or when utilizing the BFF statistic, as illustrated in Figure 7. However, across the broader range of experimental configurations, this approach generally exhibits weaker results compared to the tree-based methods.

- Appendix A.5.2 and Figure 7 further corroborate the effectiveness of our methods, showing that both `TRUST++` tuned and `TRUST++` MV perform consistently well across all statistics, with highlights for the BFF statistics. Boosting also remains a good competitor, particularly for Waldo and E-value. Notably, in most scenarios where either `TRUST++` tuned or `TRUST++` MV attains the best performance, the other method performs within a narrow margin, effectively placing both among the top-performing approaches.

Figure 8 further illustrates the confidence regions produced by each method alongside the uncertainty quantification provided by `TRUST++`. While `TRUST++`, boosting, and NFlows all closely approximate the oracle region, the Monte Carlo method significantly underestimates the set's size. Specifically, the NFlows region appears slightly more conservative than the oracle, whereas `TRUST++` and boosting provide tighter approximations. Beyond the primary region estimates, `TRUST++` distinguishes itself by offering an additional layer of uncertainty quantification. This layer reveals insights into the oracle information that may not be fully captured by the standard region estimators, likely due to constraints in the simulation budget.

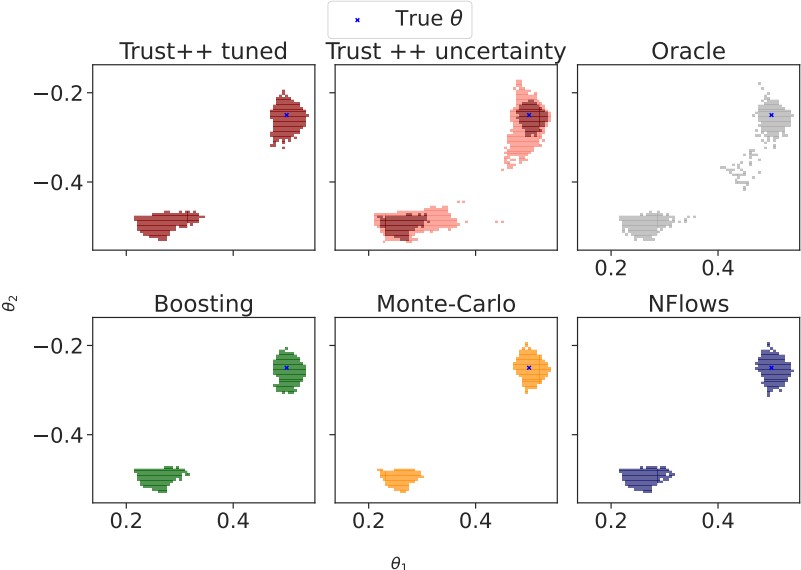

Figure 8: 95% confidence regions and `TRUST++` uncertainty for the two-moons example using the e-value statistic ($n = 20$, $B = 15{,}000$). Maroon areas denote high confidence, while salmon areas indicate uncertainty. `TRUST++` closely matches the oracle set and highlights where additional simulations could reduce uncertainty.

## 5.3 Examples of nuisance-parameter problems

In this section, we demonstrate the application of both of our methods in scenarios involving nuisance parameters. We begin with a Poisson counting experiment, originally introduced by Dalmasso et al. (2024) as an example from high-energy physics, where the BFF statistic is non-invariant, requiring both estimation and marginalization. Next, we tackle a classic inference problem: estimating parameter intervals within a gamma generalized linear model (GLM) under the presence of nuisance parameters and limited sample sizes.

We compare our methods to the standard asymptotic approach, which employs a likelihood ratio statistic marginalized over the nuisance parameter. For illustrative simplicity, we focus on the `TRUST++`-MV variant in this context. To ensure greater robustness in the more sensitive nuisance parameter regime, we set the minimum number of samples per leaf to 550. This increased constraint helps stabilize the local cutoff estimates against the additional variability introduced by the marginalized parameters.

### 5.3.1 Poisson Counting Experiment

We consider the observed data $\mathbf{X} = (N_b, N_s)$ which consists of two counts where $N_b \sim \text{Pois}(\nu\tau b)$ and $N_s \sim \text{Pois}(\nu b + \mu s)$. In this example, $b$, $s$, and $\tau$ are fixed hyperparameters, while $\mu$ is the parameter of interest and $\nu$ is the nuisance parameter. We define $\boldsymbol{\theta} = (\mu, \nu) \in \Theta = (0, 5) \times (0, 1.5)$ and fix the hyperparameters as $s = 15$, $b = 70$ and $\tau = 1$ to avoid the Gaussian limiting regime for Poisson distributions. We also take the uniform distribution over the parameter space as prior. To derive a statistic $\tau(\mathbf{X}, \mu)$ in this case, in this context, we use a marginalized BFF statistic based on the posterior $f(\mu|\mathbf{x})$, estimated through a normalizing flows approach (details in Appendix A). We set the simulation budget to $B = 10 \times 10^3$ and $n = 1$ to fit all methods.

Although the BFF statistic is marginalized over nuisance parameters, it may still depend on their values. Therefore, we fit all methods considering the full parameter $(\mu, \nu)$ and compute the cutoff for $\mu$ through Eq. 7. Our approaches facilitate this computation by using the tree structure and achieve better results, as shown in Table 1. Table 1 highlights the strong performance of `TRUST++`, with coverage closely aligning with the oracle, emphasizing its adaptability to challenges involving nuisance parameters. Also, we observe that the Monte Carlo method shows a substantial deviation from the oracle, further illustrating its limitations in adapting effectively to different inference scenarios.

Table 1: Mean absolute coverage deviation from the oracle for each method in the Poisson counting example. The average across 15 runs is reported along with twice its standard error. `TRUST++` demonstrates exceptional performance.

| Methods | $d_\alpha$ | $SE \cdot 2$ |
|---|---|---|
| TRUST | 0.0369 | $0.14 \cdot 10^{-4}$ |
| TRUST++ | 0.0041 | $0.15 \cdot 10^{-4}$ |
| Boosting | 0.0084 | $0.12 \cdot 10^{-4}$ |
| MC | 0.1460 | $0.11 \cdot 10^{-4}$ |

While `TRUST++` achieves the best results, boosting also performs well, providing coverage that is similarly close to the oracle's. Figure 9 details the differences between the methods. We observe that `TRUST++` effectively controls coverage deviation across all combinations of $(\mu, \nu)$. In contrast, boosting exhibits areas of higher deviation, particularly around $\mu = 2$ for all values of $\nu$ and around $\mu = 3$ when $\nu$ is low. This demonstrates `TRUST++`'s accuracy in approximating the oracle region throughout the parameter space, indicating its effectiveness in estimating the cutoff $C_\mu$ defined in Eq. 7.

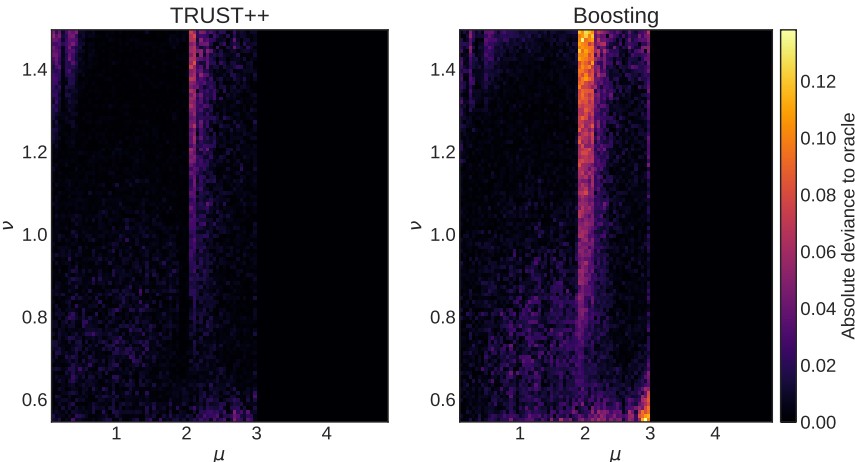

Figure 9: Comparison of absolute deviations from the oracle for each pair of parameters between `TRUST++` and boosting. Notably, around $\mu = 2$, boosting exhibits significant deviations from the oracle, whereas `TRUST++` effectively controls these deviations, keeping them below 0.1.

### 5.3.2 Gamma GLM experiment

Consider i.i.d data $Y_i \in \mathbb{R}^+$ generated from a gamma generalized linear model (GLM) with fixed covariates $\mathbf{X}_i = (X_{i,1}, X_{i,2}) \in (-1, 1)^2$:

$$Y_i \sim \text{Gamma}\left(1/\phi, \phi \cdot \exp\left\{\beta_0 + \beta_1 X_{i,1} + \beta_2 X_{i,2}\right\}\right) , \tag{13}$$

where $\phi$ is the dispersion parameter and $\beta_0, \beta_1$ and $\beta_2$ are coefficients for the linear predictor, with $\phi \in (0, 1.75)$ and $\boldsymbol{\beta} = (\beta_0, \beta_1, \beta_2) \in \mathbb{R}^3$. The expected value of each $Y_i$ is given by:

$$\mathbb{E}[Y_i] = \exp\left\{\beta_0 + \beta_1 X_1 + \beta_2 X_2\right\} .$$

A standard problem in this setting is to derive a valid confidence interval for a single parameter $\beta_j$. A widely used approach is to construct an asymptotic confidence interval based on the marginalized likelihood ratio statistic: $LR(\mathbf{y}, \beta_j) = \log \mathcal{L}(\mathbf{y}; \widehat{\boldsymbol{\beta}}_{-j}, \beta_j) - \log \mathcal{L}(\mathbf{y}; \widehat{\boldsymbol{\beta}})$, where $\widehat{\boldsymbol{\beta}}$ is the MLE, $\widehat{\boldsymbol{\beta}}_{-j}$ the MLE restricted under a fixed $\beta_j$ and $\mathcal{L}(.; \boldsymbol{\beta})$ denotes the GLM likelihood as defined in Eq. 13. Using this statistic, an asymptotic

confidence interval can be easily derived by referencing the $\chi_1^2$ distribution to compute $C_{\beta_j}$ in a invariant way.

The limitation of this approach is that it relies on large sample sizes to be effective; for small samples, it may fail to produce a valid confidence interval, as illustrated in Figure 1(b). This is because the marginalized likelihood ratio statistic still depends on the remaining nuisance parameters, $\boldsymbol{\beta}_{-j}$ and $\phi$ in low-sample regimes. This limitation can be addressed with our approach by computing $C_{\beta_j}$ using Eq. 7 in a scalable manner. To apply all simulation-based approaches, we assume the priors $\boldsymbol{\beta} \sim N(0, 4) \cdot N(0, 1)^2$ and $\phi \sim$ Truncated Exponential$(1, 1.75)$. For method comparison, we set the parameter of interest to $j = 1$, the sample size to $n = 50$ and the simulation budget to $B = 10 \cdot 10^3$, with the covariate matrix $\mathbf{X}$ fixed according to values generated independently from $U(-1, 1)^2$. Table 2 shows the mean deviation from the oracle coverage for each method, while Figure 1 illustrates the probability of coverage and confidence intervals for specific values of $\beta_j$ and $\mathbf{Y}$ samples.

Table 2: Mean absolute coverage deviation from the oracle for each method in the GLM example. The average across 15 runs is reported along with twice its standard error. `TRUST++` demonstrates superior performance compared to all competing methods.

| Methods | $d_\alpha$ | $SE \cdot 2$ |
|---|---|---|
| TRUST | 0.0312 | $0.176 \cdot 10^{-3}$ |
| TRUST++ | 0.0157 | $0.163 \cdot 10^{-3}$ |
| Boosting | 0.0234 | $0.171 \cdot 10^{-3}$ |
| MC | 0.0207 | $0.157 \cdot 10^{-3}$ |
| Asymptotic | 0.0327 | $0.171 \cdot 10^{-3}$ |

By Table 2 we notice that `TRUST++` achieves the best performance, with coverage closely matching the oracle and outperforming Monte Carlo and boosting by a large margin. Figure 1 illustrate this proximity for specific values of $\beta_j$ and observed sample values $\mathbf{Y}$. Although `TRUST++` exhibits overcoverage relative to the nominal level $1 - \alpha$ and produces larger confidence intervals compared to other methods, it closely emulates the oracle's behavior, making it more valid than the competing approaches. Additionally, `TRUST++`'s uncertainty bar indicates that for nearly all parameters within the confidence interval, we can be confident they truly lie within the interval, reflecting a well-estimated range. Furthermore, both Table 2 and Figure 1 reveal that the asymptotic approach not only deviates more from the oracle but also undercovers and underestimates the confidence interval length. This underscores the limitations of asymptotic methods for small sample sizes and nuisance parameter, even in problems with tractable likelihoods.

## 6 Final Remarks

This paper introduces novel methods, `TRUST` and `TRUST++`, for the distribution-free calibration of statistical confidence sets, ensuring both finite-sample local coverage and asymptotic conditional coverage when $M = K$. In practice, $M < K$ also led to good results. By leveraging tools from conformal inference, we create robust confidence sets that extend the applicability of traditional conformal techniques, typically used for generating prediction intervals, to the domain of statistical inference. This adaptation allows us to maintain the desired coverage properties even in challenging settings. In practice, this translates to improved performances across a variety of scenarios, particularly in cases involving small sample sizes $n$ and larger simulation budgets $B$.

Furthermore, our methods effectively address inference settings with nuisance parameters, a notable challenge for other approaches, especially in likelihood-free inference. Unlike existing techniques, `TRUST` and `TRUST++` also enable robust uncertainty quantification about the oracle confidence sets, providing valuable insights into whether additional simulated data is needed to achieve more reliable estimates. Additionally, we can leverage the third branch of LF2I (Dalmasso et al., 2020; 2024) to test for the exact conditional coverage of our confidence sets.

Theorem 3 demonstrates that both `TRUST` and `TRUST++` are consistent estimators for quantile regression. While we applied these methods specifically to quantile-regress the conformal score on $\theta$, their utility extends

far beyond this particular setting. They can be applied in a variety of contexts, such as constructing predictive intervals in a prediction setting, including for multivariate $Y$. This capability allows for the generation of label-conditional predictive sets that aim to control $\mathbb{P}(Y \in \mathbb{R}(\mathbf{X}) \mid Y = y)$, ensuring coverage conditional on the label.

While our methods are observation-efficient in the sense that they can operate with a small number of real observations, they are simulation-based and therefore depend on a simulation budget $B$. Consequently, if simulator runs are very expensive, generating the required simulations may be the dominant computational cost, as in many likelihood-free inference pipelines.

To our knowledge, no other conformal methods aim to control label-conditional coverage for continuous outcomes $Y$, making this a novel contribution to the field. This work paves the way for future research to explore label-conditional inference in complex, continuous and multivariate-label settings, potentially improving the reliability of prediction intervals in domains where conditional control is critical.

### Acknowledgments

L.M.C.C is grateful for the fellowships provided by São Paulo Research Foundation (FAPESP), grants 2022/08579-7 and 2025/06168-8. R. I. is grateful for the financial support of FAPESP (grants 2019/11321-9 and 2023/07068-1) and CNPq (grants 422705/2021-7, 305065/2023-8 and 403458/2025-0). R. B. S. produced this work as part of the activities of Fundação de Amparo à Pesquisa do Estado de São Paulo Research, Innovation and Dissemination Center for Neuromathematics (grant 2013/07699-0). The authors are also grateful to Rodrigo F. L. Lassance and Ann B. Lee for their suggestions and insightful discussions.

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

# A   Experiment details

In this section, we provide supplementary details regarding our experimental methodology and present additional results. We specify the statistics and models/simulators used for both the tractable and intractable likelihood comparisons and describe the architectures for the LFI posterior estimator and the estimator used in the Poisson Counting experiment.

For implementation, all experiments involving normalizing flows for posterior estimation were conducted using the *normflows* Python package (Stimper et al., 2023). We employed an autoregressive rational quadratic spline architecture for each flow (Durkan et al., 2019), and the models were trained on an Acer Nitro 5 AN515-54 using a GPU (CUDA).

## A.1   Statistics and models details

### A.1.1   Tractable likelihood

The data-generating processes are:

- **Normal model with fixed variance**: $X_i \overset{iid}{\sim} N(\theta, 1)$, with $\theta \in \Theta = [-5, 5]$. When a prior is needed to build $\tau$, we use $\theta \sim N(0, 0.25)$.

- **Gaussian mixture model (GMM)**: $X_i \overset{iid}{\sim} 0.5N(\theta, 1) + 0.5N(-\theta, 1)$, with $\theta \in \Theta = [0, 5]$. When a prior is needed to build $\tau$, we use $\theta \sim N(0.25, 1)$ as the prior.

- **Lognormal model with both mean and scale as parameters**: $X_i \overset{iid}{\sim} \text{lognormal}(\mu, \sigma^2)$, with $\boldsymbol{\theta} = \mu \times \sigma^2 \in [-2.5, 2.5] \times [0.15, 1.25] = \boldsymbol{\Theta}$. When a prior is needed to build $\tau$, we use $(\mu, \sigma^2) \sim$ NIG$(0, 2, 2, 1)$.

To choose $\tau$, we consider:

- **Likelihood ratio**: Considering that $\mathcal{L}(.; \theta_0)$ represents the likelihood function, the likelihood ratio statistic is given by:

$$LR(\mathbf{x}, \theta_0) = \log \frac{\mathcal{L}(\mathbf{x}; \theta_0)}{\sup_{\theta \neq \theta_0} \mathcal{L}(\mathbf{x}; \theta)} \ . \tag{14}$$

  Under regularity conditions, Wilks' theorem (Drton, 2009) implies that $-2LR(\mathbf{X}, \theta_0)|\theta = \theta_0$ has an asymptotic $\chi_1^2$ distribution, which is typically used to approximate $C_\theta$. However, regularity conditions are not always met, as in the case of the Gaussian Mixture Model (Chen & Li, 2009).

- **Kolmogorov-Smirnov statistic**: Let $F_{\theta_0}(\cdot)$ be the theoretical CDF under $\theta_0$ and $\widehat{F}_n(\cdot)$ be the empirical CDF estimated using data $\mathbf{X}$ with size $n$. The Kolmogorov-Smirnov statistic is defined as:

$$KS(\mathbf{x}, \theta_0) = \sup_{x \in \mathcal{X}} |F_{\theta_0}(x) - \widehat{F}_n(x)| = D_n \ .$$

  In this case, a classical result used to approximate $C_\theta$ is that, under the null hypothesis:

$$\sqrt{N} D_n \overset{n \to \infty}{\longrightarrow} \mathcal{K} \ ,$$

  with $\mathcal{K}$ being the Kolmogorov distribution (Marsaglia et al., 2003).

- **Bayes Factor**: Considering a prior probability $\pi$ over $\Theta$ and the comparison of the hypothesis $H_0 : \theta \in \Theta_0$ to its complement $H_1 : \theta \in \Theta_1$, the Bayes factor (BF) is given by the ratio of the marginal likelihood of both hypothesis (Kass & Raftery, 1995):

$$BF(\mathbf{x}, \Theta_0) = \frac{\mathbb{P}(\mathbf{x}|H_0)}{\mathbb{P}(\mathbf{x}|H_1)} = \frac{\int_{\Theta_0} \mathcal{L}(\mathbf{x}; \theta) d\pi_0(\theta)}{\int_{\Theta_1} \mathcal{L}(\mathbf{x}; \theta) d\pi_1(\theta)} \ , \tag{15}$$

where $\pi_0$ and $\pi_1$ represent the restrictions of $\pi$ to $\Theta_0$ and $\Theta_1$, respectively. Since our focus is on the precise hypothesis $H_0 : \theta = \theta_0$ without altering the joint distribution of the data $\mathbf{X}$ and $\theta$, we define the restrictions as follows:

$$\pi_0(\theta) = \begin{cases} 1 \text{ if } \theta = \theta_0 \\ 0 \text{ otherwise} \end{cases} \qquad \pi_1 = \begin{cases} \pi(\theta) \text{ if } \theta \neq \theta_0 \\ 0 \text{ otherwise} \end{cases}$$

and then we compute the Bayes Factor statistic as:

$$BF(\mathbf{x}, \theta_0) = \frac{\int_{\Theta_0} \mathcal{L}(\mathbf{x}; \theta) d\pi_0(\theta)}{\int_{\Theta_1} \mathcal{L}(\mathbf{x}; \theta) d\pi_1(\theta)} = \frac{\mathcal{L}(\mathbf{x}; \theta_0)}{f(\mathbf{x})} = \frac{f(\theta_0, \mathbf{x})}{\pi(\theta_0) \cdot f(\mathbf{x})} = \frac{f(\theta_0|\mathbf{x})}{\pi(\theta_0)} \ . \tag{16}$$

Following Dalmasso et al. (2024), we use the Bayes Factor as a frequentist statistic to construct confidence sets. We adopt the term "Bayes Frequentist Statistic", as introduced by Dalmasso et al. (2024).

- **E-value:** Another statistic used to test hypotheses in a Bayesian context is the Full Bayesian Significance Testing (Pereira & Stern, 1999; de B Pereira et al., 2008). This procedure assigns an evidence measure based on the posterior distribution to the hypothesis $H_0$ and rejects it if the evidence measure is small. Specifically, let $T_{\theta_0} = \{\theta \in \Theta \mid f(\theta|\mathbf{X}) \geq f(\theta_0|\mathbf{X})\}$. The Bayesian evidence measure (e-value) in favor of $H_0$ is defined as:

$$ev(\mathbf{x}, \theta_0) = 1 - \mathbb{P}(\theta \in T_{\theta_0}|\mathbf{x}) = 1 - \int_{T_{\theta_0}} f(\theta|\mathbf{x}) d\theta. \tag{17}$$

Although the e-value is originally considered a Bayesian measure, we use it as a frequentist statistic to construct confidence sets. Therefore, we use the term "Frequentist e-value" to refer to this statistic in our context. Diniz et al. (2012) establishes an asymptotic connection between the e-value and the likelihood ratio statistic, providing an asymptotic approximation for the e-value under contour restrictions. The resulting asymptotic cutoff corresponds to the significance level $\alpha$ for the simulation settings used here.

### A.1.2 Intractable likelihood

We consider the following simulators:

- **SLCP (Simple likelihood complex posterior)**: in this model, we consider that $\mathbf{x} = (\mathbf{x}_1, \mathbf{x}_2, \mathbf{x}_3, \mathbf{x}_4) \in \mathbb{R}^8$ represents 2d-coordinates of 4 points. The coordinates of each point are sampled independently from a multivariate Gaussian whose mean and covariance matrix are parametrized by a 5-dimensional parameter $\boldsymbol{\theta}$ (Hermans et al., 2022). Despite the likelihood's simplicity, the posterior of this model is complex (Papamakarios et al., 2019), making it difficult to use the BFF and E-value statistics in the analytical formula. We let $\Theta = (-3, 3)^5$ and use $\theta_i \sim U(-3, 3)$ as a prior for the model and to estimate $\tau$.

- **Two Moons**: here we consider a likelihood model with $\mathbf{x} = (x_1, x_2) \in \mathbb{R}^2$ and a two-dimensional $\boldsymbol{\theta}$ such that the posterior exhibits a bimodal moon shape-like structure (Greenberg et al., 2019). In this case, we must model and approximate the posterior to estimate the BFF and E-value statistics. We set $\Theta = (-1, 1)^2$ and use $\theta_i \sim U(-1, 1)$ as a prior for the model and to estimate $\tau$.

- **M/G/1**: this model describes a processing and arrival queue system, where a 3-dimensional parameter $\theta$ influences both the service time per customer and the intervals between arrivals (Hermans et al., 2022). Here, the sample $\mathbf{x} = (x_1, \ldots, x_5) \in \mathbb{R}^5$ consists of 5 equally spaced quantiles of inter-departure times. In this case, the likelihood $\mathcal{L}(\mathbf{x}; \theta)$ is intractable, and computing the BFF and E-value statistics through likelihood-based inference would require high-dimensional integration (Blum & François, 2010). We set $\Theta = (0, 10)^2 \times (0, 1/3)$ and use $(\theta_1, \theta_2, \theta_3) \sim U(0, 10)^2 \times U(0, 1/3)$ as prior for the model and to estimate $\tau$.

- **Weinberg**: this simulator consists of a high-energy particle collision physics model (Hermans et al., 2022). Here, our sample $\mathbf{x} \in \mathbb{R}$ is a measure of the Weinberg angle and we are interested in inferring the Fermi's constant $\theta \in \mathbb{R}$. As the likelihood is intractable, we must estimate all kinds of statistics for this problem. We set $\Theta = (0.5, 1.5)$, and use the prior distribution $\theta \sim U(0.5, 1.5)$ for estimating $\tau$.

- **SIR**: In this example, we examine an epidemiological model that tracks the dynamics of individuals across three states: susceptible (S), infectious (I), and recovered or deceased (R) (Lueckmann et al., 2021). The sample $\mathbf{x} = (x_1, x_2, x_3) \in \mathbb{R}^3$ represents the count of individuals in each state within a population of 1000, observed after 10 iterations of the model. The parameter $\boldsymbol{\theta}$, a 2-dimensional vector, denotes the contact rate and the mean recovery rate of the model. This case also involves an intractable likelihood, requiring estimation of all relevant statistics. We set $\Theta = (0, 0.5)^2$ and apply the prior $\theta_i \sim U(0, 0.5)$ for estimating $\tau$.

For choices of $\tau$, we consider both the E-value and BFF introduced in Appendix A.1.1 along with the Waldo statistic (Masserano et al., 2023):

$$\text{Waldo}(\mathbf{x}, \theta_0) = (\mathbb{E}[\theta|\mathbf{x}] - \theta_0)^T \mathbb{V}[\theta|\mathbf{x}]^{-1} (\mathbb{E}[\theta|\mathbf{x}] - \theta_0) , \tag{18}$$

where $\mathbb{E}[\theta|\mathbf{x}]$ and $\mathbb{V}[\theta|\mathbf{x}]$ replace the MLE estimator $\hat{\theta}$ and its variance by the conditional mean and covariance matrix of $\theta$ given $\mathbf{x}$. As detailed by Masserano et al. (2023), under Bayes estimator assumptions, the Waldo statistic retains the same asymptotic properties as the Wald statistic. However, for smaller $n$, Waldo can leverage consistent priors on $\theta$ to produce tighter confidence sets. This makes Waldo a compelling choice in scenarios where the likelihood is intractable.

## A.2 LFI posterior estimator

In the LFI setting, we employed a total of six flows, each consisting of two hidden layers with 128 units per layer. A dropout probability of 0.35 was applied to each layer to prevent overfitting. For optimization, we trained the model for a maximum of 1000 epochs, with early stopping triggered after 30 epochs of no improvement. The Adam optimizer was used with a learning rate of $3 \times 10^{-4}$ and a weight decay of $1 \times 10^{-5}$. To train the neural network, we simulated $B = 30,000$ samples for all sample sizes $n$.

## A.3 Normalizing flow cutoff estimation

To estimate the conditional distribution of the test statistics $H(\cdot|\theta)$ through a conditional normalizing flow, we consider the architecture consisting of a sequence of 4 flow layers, where each transformation is conditioned on the parameter vector $\theta$. The conditioning logic is processed through a neural network featuring 2 hidden layers with 64 units each.

During training, we utilize a simulation budget of $B = 10000$ and minimize the negative log-likelihood using the Adam optimizer with a batch size of 250. To prevent overfitting, we implement an early stopping criterion with a patience of 50 epochs. Once the model is fitted, the $(1 - \alpha)$-quantile $C_\theta$ is computed by generating $B_{quantile} = 2000$ samples from the learned conditional density $f(H|\theta)$ for each evaluation point in the parameter space and extracting the corresponding empirical quantile.

## A.4 Poisson Counting Experiment posterior estimator

For the nuisance parameter experiment, we used a total of four flows with the same configuration for hidden layers, hidden units, dropout probability, and optimizer as in the LFI neural posterior model. In this case, training was conducted for up to 2000 epochs, with early stopping triggered after 200 epochs without improvement. The neural network was trained using $B = 25,000$ simulated samples for $n = 1$.

## A.5 Experiment results

This subsection presents additional heatmaps comparing the conditional coverage performance of each method for each statistic separately. In each heatmap, rows correspond to the methods and columns rep-

resent specific scenarios, defined by a model and budget combination. The heatmaps are divided by each sample size. For each scenario, we highlight the top-performing methods in green, which are those with the lowest MAE or an MAE within two standard errors of the minimum.

### A.5.1 Tractable likelihood detailed results

Figures 5, 10, 11, and 12 show the performance of the BFF, LR, KS, and E-value statistics, respectively, for sample sizes of $n = 10, 20, 50, 100$.

Overall, TRUST++ tuned and TRUST are strong competitors for tractable models. Nonetheless, certain methods are preferable in specific scenarios. For instance, the asymptotic approach performs particularly well for the LR statistic under the normal model with fixed variance (Figure 10). Similarly, boosting shows competitive performance for the E-value: while not consistently superior across all setups, its performance improves with larger sample sizes. For the E-value, our methods perform well for $n = 10, 20$, but their advantage diminishes as the sample size increases.

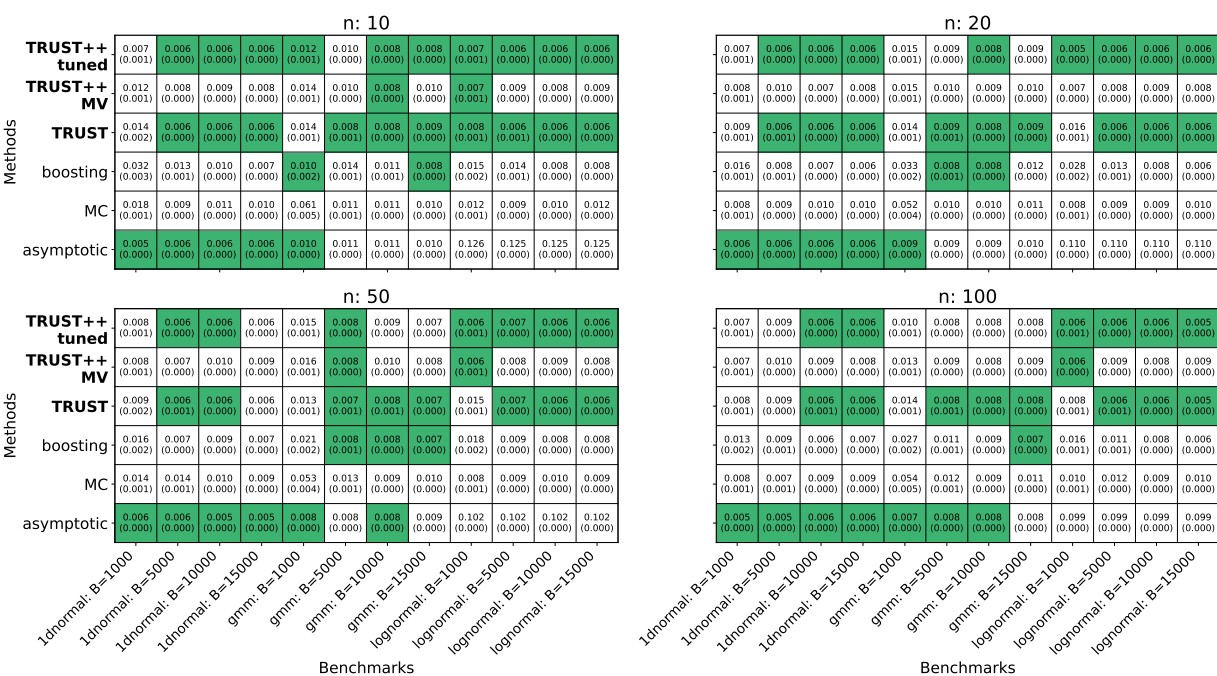

Figure 10: Best methods (lowest MAE) for LR on tractable models. The asymptotic is always among the best competitors for the Normal model with fixed variance D.G.P., and also appears in some setups for the GMM. Again, our methods are consistently good performers, especially TRUST++ tuned and TRUST.

### A.5.2 Intractable likelihood detailed results

Figures 7, 13, and 14 present the models' performance for the BFF, E-value, and Waldo statistics, respectively, in the LFI experiments with sample sizes $n = 1, 5, 10, 20$.

TRUST++ tuned, TRUST++ MV, and boosting emerge as the overall best performers. TRUST achieves reasonable results in certain scenarios (e.g., Waldo on the SIR benchmark with $n = 10$) but rarely ranks among the top methods. MC, in contrast, consistently performs poorly for intractable likelihoods.

For $n = 1$, TRUST++ tuned, TRUST++ MV, and boosting achieve strong results across all statistics. Interestingly, TRUST performs well for the E-value in the tractable generator, showing good coverage control for $B = 15,000, 20,000, 30,000$.

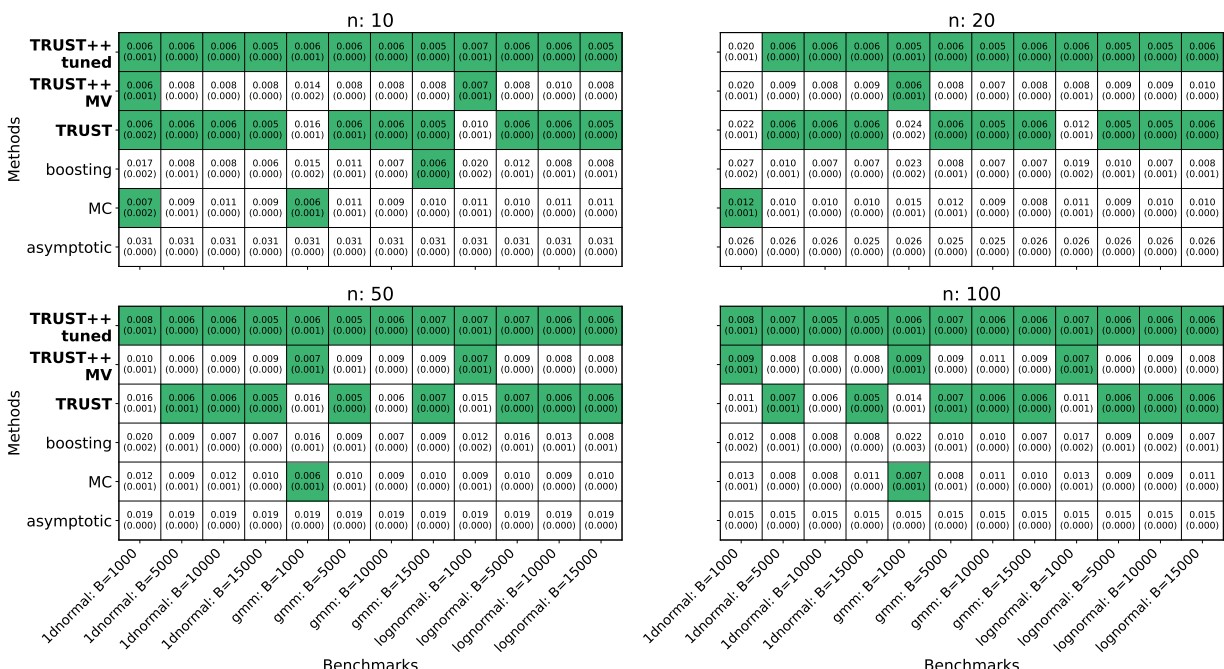

Figure 11: Best methods (lowest MAE) for KS on tractable models. `TRUST++` tuned is among the best methods in 47 out of 48 setups, proving to be a powerful competitor. `TRUST` is also very consistent, appearing in the majority os scenarios. MC and `TRUST++` MV have appeared less than the other two, but are considerably more present than boosting and asymptotic, which are not a good choice for this statistic, as shown in the graph.

## A.6 Hyperparameter sensibility

To provide a comprehensive view of the robustness of `TRUST++`, we include a sensitivity analysis in this section. For this study, we utilize the two moons simulator with a simulation budget of $B = 15000$. Across 30 independent experiment repetitions, we evaluate the influence of three core decision tree hyperparameters on the coverage MAE, while holding the others at their default values: $K = 200$ trees, `min_samples_leaf` $= 300$, and `max_depth` = None.

The configurations compared in our analysis are summarized below:

- Number of Trees ($K$): We compare $K \in \{50, 100, 200, 400, 500, 700\}$, using $K = 200$ as the base default.

- Minimum Samples per Leaf: We evaluate values $\in \{50, 100, 200, 300, 500, 750, 1000, 1250\}$, with a default of 300

- Maximum Tree Depth: We test varying depths $\in \{5, 10, 20, \text{None}\}$, where None is the default and denotes unlimited growth, being ssigned numerically to 0

- Neighborhood Parameter ($M$): Throughout these tests, $M$ is fixed at $K/2$, placing the model in a majority-vote regime.

Figure 15 displays both the coverage MAE and the computational runtime for `TRUST++` across each hyperparameter combination. The results underscore that the most critical performance factor is the `min_samples_leaf` hyperparameter. We observe that very small leaf sizes lead to a significant spike in MAE; however, once the threshold reaches approximately 300 samples, the performance plateaus. This

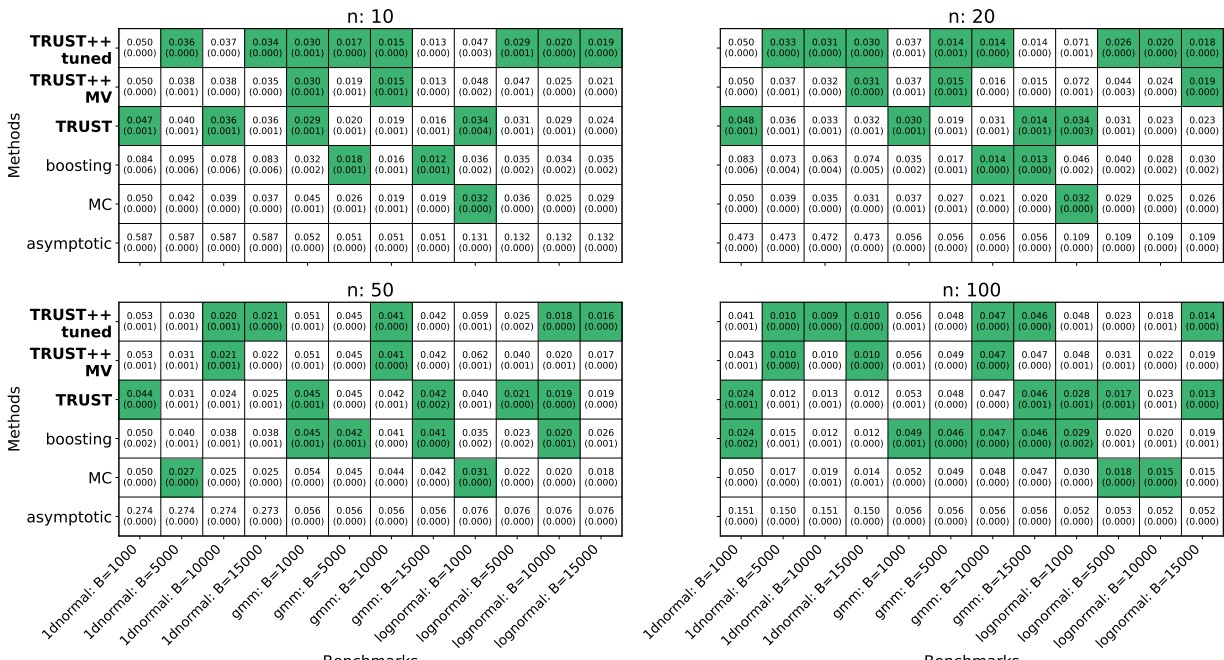

Figure 12: Best methods (lowest MAE) for E-value on tractable models. In this case, we do not see a specific method that shows a clear dominance as in Figure 11. However, it is possible to notice the poor performance of asymptotic, which is never among the best methods. Our methods and the boosting are consistent across the setups.

indicates that the local neighborhood has become sufficiently well-populated to provide reliable quantile estimates.

Conversely, the performance remains remarkably stable across all other configurations, suggesting that `TRUST++` is robust to secondary architectural choices. In terms of computational efficiency, only the number of trees $K$ demonstrates a linear increase in runtime, while MAE remains consistently low. This supports our default hyperparameter selections, demonstrating that `TRUST++` achieves convergence in its proximity measure relatively early and prioritizes the quality of the local partition over sheer ensemble volume.

### A.7   Performance under higher dimensions

To evaluate the scalability of our framework across varying parameter dimensions, we extend the Gamma GLM analysis presented in Section 5.3. Unlike the earlier nuisance parameter focus, we now consider the joint parametric space $(\phi, \beta_0, \beta_1, \ldots, \beta_p)$, where $p$ represents the increasing number of covariates.

Our benchmarking maintains a simulation budget of $B = 10000$ and utilizes the same default hyperparameters for `TRUST` and the `TRUST++`-MV variant as established in the nuisance experiments. We assess the performance of all methods across 30 experiment repetitions for dimensions $p \in \{5, 10, 50\}$. To ensure simulation stability in higher dimensions, we adapt the priors as follows:

- For $p \in \{5, 10\}$: We utilize the priors from Section 5.3.2, with $\boldsymbol{\beta} \sim N(0, 4) \times N(0, 1)^{p-1}$ and $\phi \sim$ Truncated Exponential$(1, 1.75)$.

- For $p = 50$: $\boldsymbol{\beta} \sim N(0, 0.0625) \times N(0, 0.01)^{49}$ and $\phi \sim$ Truncated Exponential$(1, 1.5)$.

The comparative performance across these dimensions, illustrated in Figure 16, reveals a strong contrast between our data-adaptive partitions and traditional benchmarks. While Monte Carlo (MC) approaches

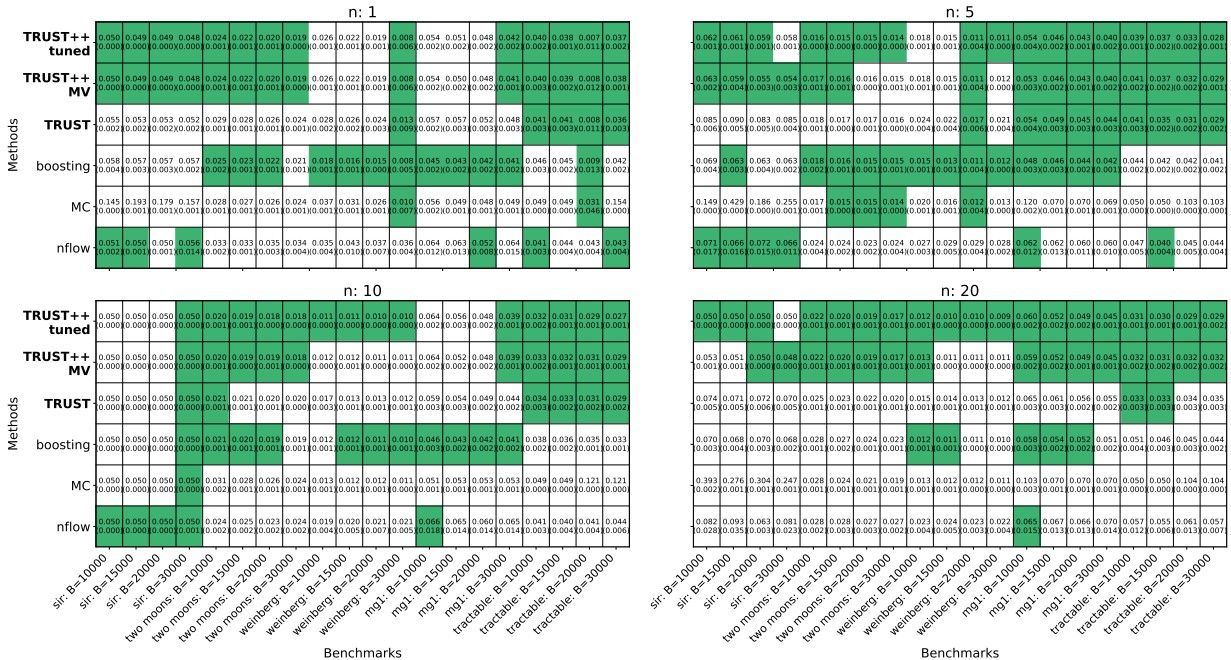

Figure 13: Best methods (lowest MAE) for E-value on intractable models. As on BFF, `TRUST++` tuned, `TRUST++` MV and boosting are the most consistent competitors. However, for this statistic, `TRUST` stands out in some scenarios, especially when $n = 5, 10$ (i.e., not in the extreme cases). As in the BFF case, nflow is competitive for the SIR benchmark with $n = 1, 5, 10$.

fail to scale—becoming computationally prohibitive and eventually failing to run at $p = 50$ due to memory constraints imposed by the massive grid size—both `TRUST` and `TRUST++` maintain consistently low coverage MAE as dimensionality increases.

Furthermore, while asymptotic approximations exhibit a clear degradation in performance as $p$ grows, our tree-based methods effectively mitigate the "curse of dimensionality" by focusing on the most informative regions of the parameter space. This targeted partitioning keeps the MAE nearly flat between $p = 10$ and $p = 50$. Ultimately, both `TRUST` variants remain competitive with the boosting-based alternative across the entire spectrum, demonstrating that our local neighborhood logic remains valid and robust even in high-dimensional settings where traditional approximations falter.

## B   Algorithm for tuning $M$

In Algorithm 3, we specify the procedure for tuning $M$ by using an available validation grid.

## C   Proofs

### C.1   Section 2 - Methodology

*Proof of Theorem 1.* Notice that

$$\mathbb{P}\left(\theta \in \mathcal{I}(\mathbf{x}) | \theta \notin R(\mathbf{x})\right) = \mathbb{P}\left(\tau(\mathbf{x}, \theta) \geq \widehat{C}_\theta^U | \tau(\mathbf{x}, \theta) \leq C_\theta\right)$$

$$= \mathbb{P}\left(\widehat{C}_\theta^U \leq C_\theta | \tau(\mathbf{x}, \theta) \leq C_\theta\right) \leq \beta/2,$$

which proves the first statement. The proof for the second statement is analogous. □

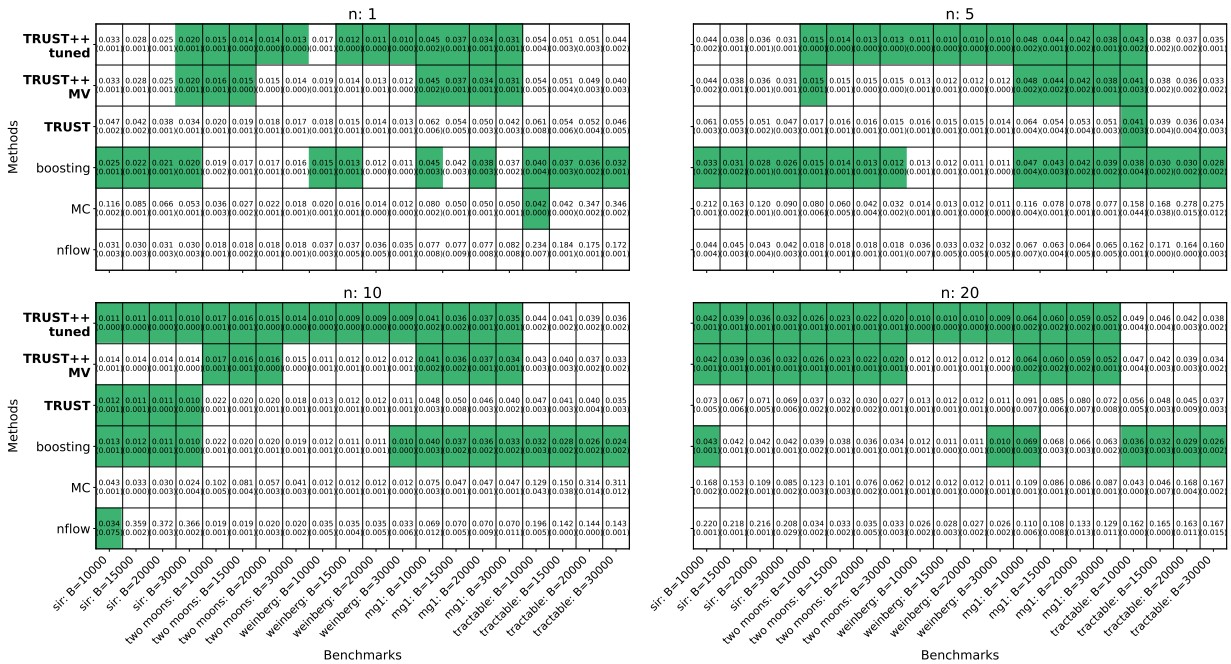

Figure 14: Best methods (lowest MAE) for Waldo on intractable models. Similarly to BFF, `TRUST++` tuned, `TRUST++` MV and boosting are the methods that stand out the most. MC and nflow only appear as the winners once each in 80 experiments, and `TRUST` has a modest performance, with good results for the SIR benchmark with $n = 10$.

---

**Algorithm 3:** Tuning algorithm for $M$

**Data:** Validation grid size $B_{\text{tune}}$; grid of $M$'s between 0 to $K$ $M_{\text{grid}}$; fitted `TRUST++`; number of simulated statistics $n_{\text{sim}}$ in each grid point

**Result:** Tuned value of $M$, $M_{\text{tuned}}$

simulate $\Theta_{\text{grid}} = (\theta_1, \ldots, \theta_{B_{\text{tune}}})$ from the prior ;

**for** $M_{cand} \in M_{grid}$ **do**

    compute $\widehat{C}_\theta$ in `TRUST++` with $M$ fixed as $M_{\text{cand}}$ for each $\theta \in \Theta_{\text{grid}}$ ;

    compute $\text{cover}_\alpha(\widehat{C}, \theta)$ for each $\theta \in \Theta_{\text{grid}}$ (as in Eq. 10);

    $MAE_{M_{\text{cand}}} \leftarrow MAE(\widehat{C}, \alpha)$ (as in Eq. 11) ;

**end**

$M_{\text{tuned}} \leftarrow \underset{M_{\text{cand}} \in M_{\text{grid}}}{\arg\min} \{MAE_{M_{\text{cand}}}\}$ ;

**return** $M_{tuned}$

---

## C.2 Section 3 - Theoretical Results

To summarize, in `TRUST`, the partition is created by building a regression tree that uses $\theta$ as the input and $\tau(\mathbf{x}, \theta)$ as the output. This tree, trained on the simulated data

$$(\theta_1, \tau(\mathbf{X}_1, \theta_1)), \ldots, (\theta_B, \tau(\mathbf{X}_B, \theta_B)),$$

naturally induces a partition on $\Theta$.

For `TRUST++`, we extend this approach by generating $K$ regression trees. We then use Breiman's proximity measure $\rho(\theta', \theta)$, which counts the number of times $\theta'$ and $\theta$ appear together in the same leaf across these $K$ trees. Finally, we define the partition $\mathcal{A}$ based on this proximity measure, using the equivalence relation

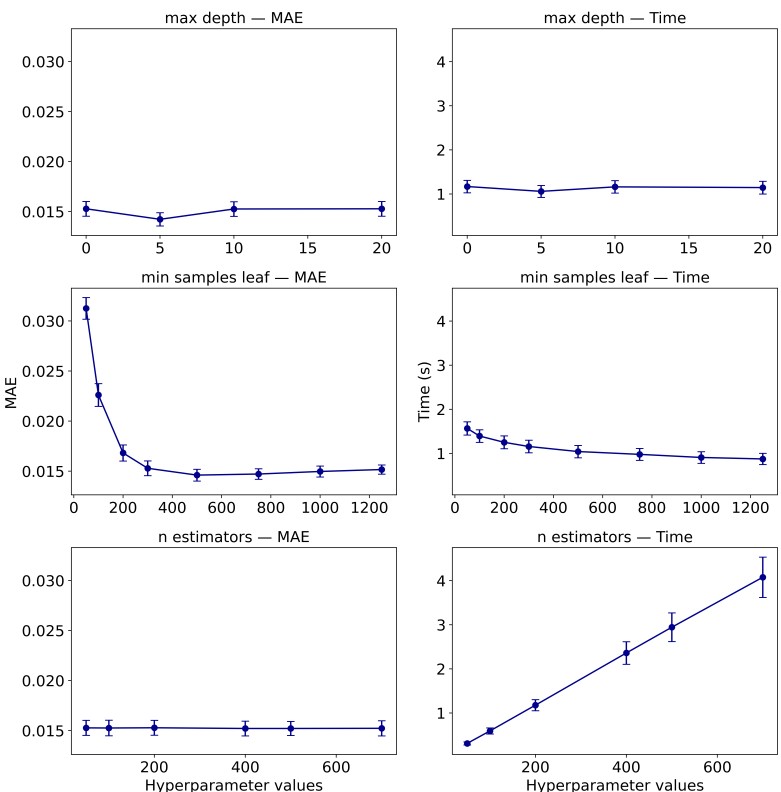

Figure 15: Sensitivity analysis of TRUST++ on the *two moons* task ($B = 15000$).The left column shows coverage MAE; the right column shows runtime (s). Each row varies one hyperparameter while others remain at defaults ($K = 200, M = K/2, \text{leaf\_size} = 300$). Results represent 30 repetitions with standard deviation error bars. Performance is primarily driven by min_samples_leaf; once leaves are sufficiently populated ($\approx 300$), coverage remains stable and invariant to tree depth or ensemble size, though the latter increases runtime linearly.

$\theta \sim \theta' \iff \rho(\theta', \theta) = K$; in other words, $\theta$ and $\theta'$ must appear together in the same leaves across all $K$ trees.

This framework ensures that the partition $\mathcal{A}$ meets the necessary structural conditions to apply conformal prediction as stated in Theorem 2.

*Proof of Theorem 2.* This is a straightforward application of conformal prediction (see Angelopoulos et al. (2023)). The complete proof can be found in detail in (Cabezas et al., 2025b, Theorem 2). □

To prove Theorem 3 we need the following lemma.

**Lemma 1.** *Let $H(t|\theta) = \mathbb{P}(\tau(\mathbf{X}, \theta) \le t|\theta)$ and $A$ be a measurable set. Suppose $p(x, \theta)$ is the joint PDF of $(\mathbf{X}, \theta)$, satisfying $\int p(x, \theta)\, dx = r(\theta) > 0$. Then we have*

$$\mathbb{E}[H(t|\theta)|\theta \in A] = \mathbb{P}(\tau(\mathbf{X}, \theta) \le t|\theta \in A).$$

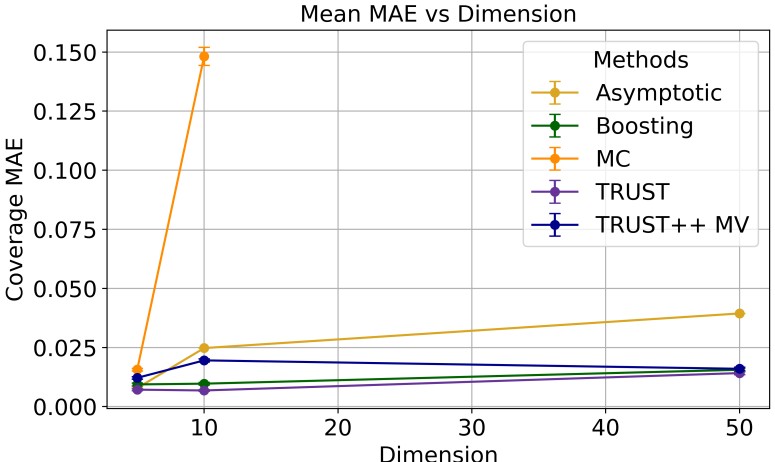

Figure 16: Scalability across dimensions $p \in \{5, 10, 50\}$. Comparison of coverage MAE for the Gamma GLM joint parameter space. While Monte Carlo (MC) becomes computationally prohibitive and fails at $p = 50$, and asymptotic methods show degrading accuracy, both TRUST and TRUST++ remain robust with consistently low MAE.

*Proof.* Define $c := \mathbb{P}(\theta \in A)$, the result follows from a straightforward calculation, as shown next.

$$
\begin{aligned}
\mathbb{E}[H(t|\theta)|\theta \in A] &= \frac{1}{\mathbb{P}(\theta \in A)} \mathbb{E}[H(t|\theta)\mathbb{I}[\theta \in A]] \\
&= c^{-1} \int H(t|\theta)\mathbb{I}[\theta \in A]r(\theta)d\theta \\
&= c^{-1} \int \mathbb{P}(\tau(\mathbf{X}, \theta) \leq t|\theta)\mathbb{I}[\theta \in A]r(\theta)d\theta \\
&= c^{-1} \int \left( \frac{1}{r(\theta)} \int \mathbb{I}[\tau(x, \theta) \leq t]p(x, \theta)dx \right) \mathbb{I}[\theta \in A]r(\theta)d\theta \\
&= c^{-1} \int \int \mathbb{I}[\tau(x, \theta) \leq t]\mathbb{I}[\theta \in A]p(x, \theta)dxd\theta \\
&= c^{-1}\mathbb{P}(\tau(\mathbf{X}, \theta) \leq t, \theta \in A) \\
&= \mathbb{P}(\tau(\mathbf{X}, \theta) \leq t|\theta \in A).
\end{aligned}
$$

$\square$

Under the Assumption 1, for a sufficiently large $B$, we achieve $H(t|\theta') \approx \widehat{H}_B(t|\theta')$, for any $\theta' \in \Theta$ and $t \in \mathbb{R}$. Based on this approximation, we outline here the intuition for how Theorem 3 will be established in detail later.

Specifically, for a fixed $\theta$ of interest, by Lemma 1:

$$
\begin{aligned}
\mathbb{P}(\tau(\mathbf{X}, \theta') \leq t \mid \theta' \in A(\theta)) &= \mathbb{E}[H(t|\theta') \mid \theta' \in A(\theta)] \\
&\approx \mathbb{E}[\widehat{H}_B(t|\theta') \mid \theta' \in A(\theta)],
\end{aligned}
$$

where the first equality results from a straightforward calculation, with its proof provided in the appendix.

By the construction of the partitions in TRUST and TRUST++, if $\theta' \in A(\theta)$, then $\widehat{H}_B(t|\theta') = \widehat{H}_B(t|\theta)$. Therefore,

$$
\begin{aligned}
\mathbb{P}(\tau(\mathbf{X},\theta') \leq t \,|\, \theta' \in A(\theta)) &\approx \mathbb{E}[\widehat{H}_B(t|\theta') \,|\, \theta' \in A(\theta)] \\
&= \mathbb{E}[\widehat{H}_B(t|\theta) \,|\, \theta' \in A(\theta)] \\
&\approx \mathbb{E}[H(t|\theta) \,|\, \theta' \in A(\theta)] \\
&= H(t|\theta).
\end{aligned}
$$

This implies that, as long as our approximation $\widehat{H}_B$ closely matches $H$, our partition-based estimate will serve as a reliable approximation of the test statistics distribution. In particular, as discussed in Section 2.1.1, let $\widehat{C}_{\theta,B} = \widehat{H}_B^{-1}(\alpha|\theta)$ denote the adjusted $\alpha$-quantile of the values $\{\tau(\mathbf{X}_b,\theta_b) : b \in I_{A(\theta)}\}$. Then,

$$
\mathbb{P}\left(\theta \notin \widehat{R}_B(\mathbf{X})|\theta\right) = H(\widehat{C}_{\theta,B}|\theta) \approx \mathbb{P}(\tau(\mathbf{X},\theta') \leq \widehat{C}_{\theta,B} \,|\, \theta' \in A(\theta)) = \alpha,
$$

suggesting that our methods indeed should achieve optimal coverage.

**Remark.** As described in Section 2, in practice we may use relaxed neighborhoods controlled by $M$. When $M < K$, these neighborhoods need not induce a partition of $\Theta$. In contrast, the asymptotic result proved here is stated under the partitioning regime $M = K$. This assumption is natural when letting the simulation budget grow ($B \to \infty$): with sufficiently many simulations, each cell is populated by enough points so that the resulting partition is non-degenerate. Accordingly, the proof of Theorem 3 below is carried out in the partitioning case $M = K$.

*Proof of Theorem 3.* Fix $\theta$. By Assumption 1, for any $\varepsilon, \delta > 0$, there exists $B_0$ and a subset $\Gamma \subset (\Theta \times \mathcal{X})^B$ such that $\mathbb{P}(\Gamma) \geq 1 - \delta$ and, conditionally on $\Gamma$,

$$
\sup_{\tilde{t} \in \mathbb{R}, \tilde{\theta} \in \Theta} \left| \widehat{H}_B(\tilde{t}|\tilde{\theta}) - H(\tilde{t}|\tilde{\theta}) \right| \leq \varepsilon.
$$

By Lemma 1,

$$
\mathbb{P}(\tau(\mathbf{X},\theta') \leq t|\theta' \in A(\theta)) = \mathbb{E}[H(t|\theta') \,|\, \theta' \in A(\theta)].
$$

Conditionally on $\Gamma$, we know that

$$
H(t|\theta') \geq \widehat{H}_B(t|\theta') - \varepsilon
$$

holds uniformly for any possible value of $\theta' \in \Theta, t \in \mathbb{R}$. Thus,

$$
\begin{aligned}
\mathbb{P}(\tau(\mathbf{X},\theta') \leq t|\theta' \in A(\theta), \Gamma) &= \mathbb{E}[H(t|\theta') \,|\, \theta' \in A(\theta), \Gamma] \\
&\geq \mathbb{E}[\widehat{H}_B(t|\theta') - \varepsilon \,|\, \theta' \in A(\theta), \Gamma] \\
&= \mathbb{E}[\widehat{H}_B(t|\theta') \,|\, \theta' \in A(\theta), \Gamma] - \varepsilon.
\end{aligned}
$$

Note that, in the partitioning regime $M = K$, the collection $\{A(\theta)\}_{\theta \in \Theta}$ forms a partition of $\Theta$. Thus, if $\theta' \in A(\theta)$, then $\theta$ and $\theta'$ belong to the same cell, implying $A(\theta') = A(\theta)$ and consequently $\widehat{H}_B(t \,|\, \theta') = \widehat{H}_B(t \,|\, \theta)$. Therefore,

$$
\begin{aligned}
\mathbb{P}(\tau(\mathbf{X},\theta') \leq t|\theta' \in A(\theta), \Gamma) &\geq \mathbb{E}[\widehat{H}_B(t|\theta) \,|\, \theta' \in A(\theta), \Gamma] - \varepsilon \\
&= \mathbb{E}[\widehat{H}_B(t|\theta) \,|\, \theta' \in A(\theta), \Gamma] - \varepsilon.
\end{aligned}
$$

Using the fact that, conditionally on $\Gamma$,

$$
\widehat{H}_B(t|\theta) \geq H(t|\theta) - \varepsilon,
$$

we obtain

$$
\begin{aligned}
\mathbb{P}(\tau(\mathbf{X},\theta') \leq t|\theta' \in A(\theta), \Gamma) &\geq \mathbb{E}[\widehat{H}_B(t|\theta) \,|\, \theta' \in A(\theta), \Gamma] - \varepsilon \\
&\geq \mathbb{E}[H(t|\theta) \,|\, \theta' \in A(\theta), \Gamma] - 2\varepsilon.
\end{aligned}
$$

Since $H(t|\theta)$ is constant, it follows that

$$\begin{aligned}
\mathbb{P}(\tau(\mathbf{X}, \theta') \leq t | \theta' \in A(\theta), \Gamma) &\geq \mathbb{E}[H(t|\theta) \,|\, \theta' \in A(\theta), \Gamma] - 2\varepsilon \\
&= H(t|\theta)\mathbb{E}[1 \,|\, \theta' \in A(\theta), \Gamma] - 2\varepsilon \\
&= H(t|\theta) - 2\varepsilon.
\end{aligned}$$

Thus, for any $t \in \mathbb{R}$:

$$\begin{aligned}
\mathbb{P}(\tau(\mathbf{X}, \theta') \leq t | \theta' \in A(\theta)) &\geq \mathbb{P}(\tau(\mathbf{X}, \theta') \leq t | \theta' \in A(\theta), \Gamma)\mathbb{P}(\Gamma) \\
&\geq (1 - \delta)(H(t|\theta) - 2\varepsilon) \\
&= H(t|\theta) - \delta H(t|\theta) - 2(1 - \delta)\varepsilon \\
&\geq H(t|\theta) - \delta - 2\varepsilon + 2\delta\varepsilon \\
&\geq H(t|\theta) - \delta - 2\varepsilon.
\end{aligned}$$

In a similar manner, we can show that

$$\mathbb{P}(\tau(\mathbf{X}, \theta') \leq t | \theta' \in A(\theta), \Gamma) \leq H(t|\theta) + 2\varepsilon.$$

Following (Meinshausen & Ridgeway, 2006, Assumption 3), we assume there exists $0 < \gamma < 0.5$ such that $\mathbb{P}(\theta' \in A(\theta)) \geq \gamma B$. This assumption is reasonable, as it implies that the partitions (i.e., the leaves) generated by the tree-based estimators are uniformly well-populated. In other words, each leaf contains a sufficient number of observations, ensuring that the coverage properties hold consistently across the partitions. Therefore,

$$\begin{aligned}
\mathbb{P}(\tau(\mathbf{X}, \theta') \leq t | \theta' \in A(\theta)) &\leq \mathbb{P}(\tau(\mathbf{X}, \theta') \leq t | \theta' \in A(\theta), \Gamma)\mathbb{P}(\Gamma) + \frac{\delta}{\gamma B} \\
&\leq \mathbb{P}(\tau(\mathbf{X}, \theta') \leq t | \theta' \in A(\theta), \Gamma) + \delta \\
&\leq H(t|\theta) + 2\varepsilon + \delta.
\end{aligned}$$

Thus, for any $\varepsilon, \delta > 0$ and any $t \in \mathbb{R}$:

$$|\mathbb{P}(\tau(\mathbf{X}, \theta') \leq t | \theta' \in A(\theta)) - H(t|\theta)| \leq 2\varepsilon + \delta.$$

Taking $t = \widehat{C}_{\theta, B}(1 - \alpha)$, by the construction of the empirical quantile over $A(\theta)$,

$$|(1 - \alpha) - H(\widehat{C}_{\theta, B}(1 - \alpha)|\theta)| \leq 2\varepsilon + \delta,$$

which is equivalent to

$$\left|(1 - \alpha) - \mathbb{P}\left(\theta \in \widehat{R}_B(\mathbf{X})|\theta\right)\right| \leq 2\varepsilon + \delta.$$

Since we can make $\varepsilon$ and $\delta$ arbitrarily small by choosing $B$ large enough, we conclude that

$$\lim_{B \to \infty} \mathbb{P}\left(\theta \in \widehat{R}_B(\mathbf{X})|\theta\right) = 1 - \alpha.$$

$\square$

