# OpenReview forum: "Conformal Calibration of Statistical Confidence Sets"
_TMLR — Accepted by TMLR_

### Review · Reviewer_nnB2 · 2025-11-17

**Summary Of Contributions:**

The paper proposes TRUST and TRUST++, two tree-based methods for estimating the distribution of a test statistic in likelihood-free settings in order to construct confidence sets with valid conditional coverage. TRUST partitions the parameter space using a regression tree, while TRUST++ uses bagged trees to obtain smoother, more accurate local neighborhoods for estimating quantiles. The paper also extends the methodology to handle nuisance parameters and provides uncertainty quantification for the resulting confidence sets. The authors provide theoretical guarantees for finite-sample and asymptotic coverage and show strong empirical performance across simulated and real-world statistical models.

**Audience:**

Yes

**Audience Explanation:**

The paper tackles a problem that appears in many simulation-based or likelihood-free inference settings, and the proposed methods would likely interest researchers working on conformal prediction, uncertainty quantification, or approximate Bayesian computation. The tree-based approach and the handling of nuisance parameters are practical ideas that could be useful to others in the community.

**Claims And Evidence:**

Yes

**Claims Explanation:**

The paper’s main claims are generally supported by the material provided. The authors give clear definitions of the problem, explain how TRUST and TRUST++ are constructed, and provide theoretical arguments for the coverage properties they aim to achieve. The experiments are reasonably designed and illustrate that the methods behave as claimed in several settings. Overall, the combination of theory and empirical results is sufficient to support the main points of the submission.

**Requested Changes:**

1- Some steps in the construction of TRUST and TRUST++, especially the motivation for the partitions and the role of $\rho$, are difficult to follow on first reading. A bit more intuition or a small running example would help readers understand the flow of the algorithm.

2- The current presentation nuisance-parameter is technically correct but hard to digest. It would help if the authors added a short summary of the key idea and gave a simple illustration of how the minimization over ν becomes tractable.

3- Since TRUST++ relies on bagging and proximity, a clearer explanation of how it differs from a regular RF-based estimator would strengthen the contribution.

4- A bit more information on simulation settings (how the $\theta$ values were chosen, how sensitive results are to tree depth or number of trees) would improve the reproducibility of the experiments.

5- Both TRUST and TRUST++ rely on tree depth, leaf size, and number of trees. The paper briefly mentions these choices, but does not fully explore how sensitive the conclusions are to these hyperparameters.

---

> ### Author Response · Authors · 2026-03-03
>
> We thank the referee for the constructive and detailed feedback. Below is our point-by-point response.
>
> - [Improved description of the methods]
>
> Thank you for this suggestion. We agree the presentation can be made more intuitive. We rewrote both the TRUST and TRUST++ subsections to add additional intuition for the partitioning step and to clarify the role of \rho (and M). We hope things are more clear now.
>
> - [Nuisance Parameters]
>
> We agree with the referee that the original version of this section was difficult to follow. We have therefore completely rewritten it for clarity and, in addition, included a new example to illustrate why the infimum becomes a minimum when restricting to trees.
>
> - [TRUST++ vs RF]
>
>
> Thank you for the suggestion. We agree that, since TRUST++ leverages bagging and a forest-style similarity notion, it is important to clarify how it differs from a standard RF-based estimator. We revised the beginning of the TRUST++ subsection to make this distinction explicit. In short, TRUST++ does not estimate $C_\theta$ by aggregating per-tree quantile predictions (e.g., averaging tree-wise cutoffs or averaging leaf-wise empirical CDFs across trees). As noted in the new comment, “averaging the $\widehat C_{\theta,B}$ values obtained across trees does not ensure distribution-free guarantees (Cabezas et al., 2025),” such aggregation generally breaks the partition-based conformal validity argument because the final cutoff is no longer an adjusted order statistic computed within a single fixed cell.
>
> - [Improved description of simulation settings and sensitivity to hiperparameters]
>
> Thanks for the suggestion. We have added a new experiment/figure to Appendix A.5, and discuss it in Section 4. The figure indicates that the minimum samples per leaf is by far the most influential hyperparameter, while performance is largely robust to the remaining settings. We also added text at the start of Section 4 (and in a few other places) explaining how each hyperparameter was chosen in practice. We hope this addresses your concern.

---

### Review · Reviewer_sUbn · 2025-11-27

**Summary Of Contributions:**

This paper addresses a critical intersection in modern statistics: the inability of traditional methods to construct valid confidence sets when sample sizes are small or when likelihood functions are intractable (Likelihood-Free Inference or LFI). The authors propose two novel methods, TRUST and TRUST++, which adapt conformal prediction techniques to statistical inference. These methods utilize simulated data to calibrate confidence sets, ensuring they achieve the correct coverage levels (e.g., 95%) even in complex, non-asymptotic settings.

Standard frequentist confidence sets rely on asymptotic theory (large sample sizes) or tractable likelihoods. In modern science (e.g., high-energy physics, epidemiology), models often define a simulator rather than a clear likelihood function.

1. Research Challenge: When $n$ (sample size) is small, or the model is complex, traditional confidence sets often under-cover or over-cover the true parameter $\theta$.

2. Research Goal: Construct a confidence set $R(X)$ such that $P(\theta \in R(X)|\theta) = 1-\alpha$ (conditional coverage) without relying on asymptotics.

The paper introduces a "distribution-free" calibration approach that bridges conformal prediction and parameter inference.

1. TRUST: A partition-based method that estimates the distribution of test statistics (like the Bayes Factor or E-value) using a tree structure.

2. TRUST++: An enhancement of TRUST that utilizes random forests (ensembles of trees) to reduce variance and improve the estimation of the critical values required for calibration.

Both methods rely on having access to a simulator (generative model) to generate training data for calibration. The paper claims to provide both finite-sample local coverage and asymptotic conditional coverage. This is significant because most LFI methods only offer approximate guarantees.

**Audience:**

Yes

**Audience Explanation:**

Statistical inference should be an important part of machine learning, while conformal prediction is one of the most interesting problems in this area.

**Broader Impact Concerns:**

May promote the research in AI4Science area.

**Claims And Evidence:**

Yes

**Claims Explanation:**

The authors tested the methods on both Tractable (Normal, GMM) and Intractable (SLCP, Two Moons, SIR, Weinberg) models.
1. Performance: TRUST++ (specifically the "tuned" and "MV" variants) consistently outperformed the baseline methods (Naive Monte Carlo and Asymptotics).
2. Robustness: In intractable likelihood settings (LFI), the naive MC method performed poorly, whereas TRUST++ maintained coverage close to the nominal level.
3. Efficiency: The "Boosting" baseline was competitive in some areas, but TRUST++ generally provided the most robust results across different sample sizes.

The methods provide uncertainty quantification for the estimated confidence sets, allowing users to know if they need more simulations to trust the result. It offers a practical solution for scientists working with "Black Box" simulators who need rigorous error bars (confidence sets) but cannot rely on standard textbook formulas due to small data limits.

**Requested Changes:**

1. Like many LFI methods, this approach relies on the ability to generate large amounts of simulated data ($B = 30,000$ in their experiments). If the simulator is extremely expensive computationally (e.g., complex fluid dynamics), this might be a bottleneck. 3. The author claimed this method as  "data efficient" (requires few real observations) but "simulation expensive".


2. While they demonstrate success on multi-dimensional parameters (e.g., 5D in SLCP), conformal methods can sometimes struggle with the "curse of dimensionality" regarding the efficiency of the partition space, though the use of Forests (TRUST++) attempts to mitigate this.


3. The experiments are restricted to relatively low-dimensional parameter spaces. The most complex model shown (SLCP) has only 5 parameters ($\theta \in \mathbb{R}^5$). Partition-based methods (like the decision trees used in TRUST) and even random forests are notoriously susceptible to the "curse of dimensionality". As the dimension of $\theta$ increases, the number of partitions required to accurately estimate the distribution grows exponentially. It is unclear if TRUST++ would maintain its performance or coverage guarantees for models with 50, 100, or more parameters, which are common in modern machine learning-based inference.

4. The baselines might be considered "weak" or "standard" rather than state-of-the-art, such as naive Monte Carlo. The paper does not appear to compare against other modern neural density estimation techniques (like Normalizing Flows used directly for calibration) or other recent conformal inference methods specifically designed for regression/conditional density estimation (e.g., CQR, DCP) adapted for this setting.

---

> ### Author Response · Authors · 2026-03-03
>
> We thank the referee for the constructive feedback. Below is our point-by-point review.
>
> - [Data efficiency]
>
> The referee is correct: our notion of “data efficiency” refers to requiring few real observations, not to requiring few simulations. We have revised the manuscript to make this distinction explicit (including in the abstract), where we now state that our methods ensure finite-sample local coverage and asymptotic conditional coverage as the number of simulations B increases, even when the observed sample size n is small. We also added a brief computational discussion to the final remarks noting that, as in many LFI methods, the approach can be bottlenecked by simulation cost when the simulator is very expensive.
>
> - [Higher Dimensions]
>
> Thank you for this observation. We agree that increasing the parameter dimension makes the problem harder for our method as well. We added a new experiment (Section 5 and Appendix A.7), which confirms this: the MAE of TRUST and TRUST++ MV increases as the dimension grows. That said, the deterioration is much more severe for the grid-based MC baseline. In fact, MC already performs much worse at 10 dimensions, and in our setup it was not practical to apply it at 50 dimensions at all (it would require at least 2^50 grid points for a fair comparison). By contrast, TRUST and TRUST++ MV remain usable and still maintain relatively small errors at 50 dimensions. Thus, while TRUST++ does not avoid the effects of increasing dimension, it degrades much more gracefully than standard MC as dimension increases.
>
> - [Other Baselines]
>
> Thanks for the suggestion. We have added Normalizing Flows to the experiments in Section 5; in our setting, they underperform both boosting and TRUST. Also, TRUST can be viewed as an adaptation of conditional predictive conformal distributions to statistical inference; we now better describe this connection in the revised manuscript (Section 2.1.1).

---

### Review · Reviewer_jmdt · 2026-02-22

**Summary Of Contributions:**

The paper introduces TRUST and TRUST++, two methods for calibrating statistical confidence sets using simulated data. The key idea for the two methods respectively is to partition the parameter space either via regression trees (TRUST) or random forests (TRUST++), then locally calibrate cutoffs using conformal ideas on simulated data. The main theoretical result (Theorem 3) establishes "B"-asymptotic (i.e., in terms of the number of simulations) conditional coverage as the simulation budget grows, which is interestingly independent of the observed sample size n. Decent experimentation was done across tractable likelihoods, LFI benchmarks, and nuisance-parameter settings show tuned TRUST++ matching or outperforming boosting, Monte Carlo, and asymptotic baselines, particularly in small-n regimes.

**Audience:**

Yes

**Audience Explanation:**

Yes. The paper should interest SBI/LFI practitioners directly and researchers interested in conformal methods. I personally found the findings interesting.

**Claims And Evidence:**

Yes

**Claims Explanation:**

This paper was well-written and well-executed.

Three main claims:
(1) Probably the main claim "TRUST/TRUST++ achieve distribution-free finite-sample local coverage and $B$-asymptotic conditional coverage" - with the main theorem for this being theorem 3. The result seems interesting, and the proof and justification looks decent to my eyes. But perhaps one thing to flag, the theorem 3 is for the strict $M = K$ variant (which induces a proper partition), whereas the generalised version with tuned $M < K$ does not always control coverage. I think the authors note this themselves on page 8 "does not always control coverage for M < K (Guan, 2023), since it does not induce a partition, in practice it has good performance", I think perhaps - if TRUST++ does not actually meet the main assumption for theorem 3 - it could be mentioned/highlighted more clearly in the paper. That said - I am not fussy on this and am happy with decent empirical results.


(2) Strong empricial results for TRUST and TRUST++. The turned TRUST++ method outperforms existing alternatives, particularly in small-n scenarios, and the comparisons seem fair. That said, the bar charts (Figures 4-5) perhaps overstate the practical advantage, and the appendix heatmaps reveal that MAE improvements over boosting are somewhat modest (but still non-negligible). The "frequency with which each method achieved the best performance" plots don't really convey this nuance. Personally, I found the appendix plots much nicer for assessing the experiments and largely ignored figures 4-5.


(3) Nuisance parameters and uncertainty quantification. No issue here to my eyes.

**Requested Changes:**

Again I think this a fairly strong paper suitable for TMLR.

Following are largely optional at author's discretion:

1. Clarify the theory-practice gap for TRUST++ (tuned): as noted before, a short remark perhaps in Section 2.2 or 3.2 explicitly stating that Theorem 3's guarantee applies to the $M = K$ partition variant, and that the tuned version operates outside this, would help. Perhaps just one sentence somewhere to make this more clear to the reader (if it really does violate the assumption for theorem 3).

2. Supplement win-count figures: figures 4–5 lose information about the magnitude of MAE differences. Personally, I'd prefer a complementary figure closer to the appendix heatmaps in the main text. But If other reviewers are fine with the win counts as-is I won't insist - and can be done at your discretion.

3. Minor typo: on page 16, "folllowing" -> "following".

---

> ### Author Response · Authors · 2026-03-03
>
> We thank the referee for the helpful comments and suggestions. Below is our point-by-point response.
>
> - [Theory vs Practice Gap]
>
> Thank you for this suggestion. We agree that making the scope of Theorem 3 explicit improves clarity. We have added a remark clarifying that Theorem 3’s finite-sample guarantee applies to the partition variant, while the tuned/relaxed TRUST++ variant (controlled by M) may operate outside the partition assumption when M<K. We also argue that in the asymptotic regime it is natural to consider M=K, since the partition elements are then expected to be sufficiently populated and hence non-degenerate.
>
> -  [Figures]
>
> We agree with the referee. We have moved some of the figures to the main text and added a remark about the magnitude of improvement compared to boosting.
>
> Also, thanks for catching the typo!

---

### Comment · Action_Editor_QLk7 · 2026-02-22
**Discussion Period**

The paper has been reviewed by 3 expert reviewers. Generally speaking, the reviewers found the paper and its contribution interesting. I invite you to respond to the reviewers and to revise the manuscript according to some of their requested changes. In particular:

  * Please take care to address Point 1 by suBN about the the simulation step being a potential bottleneck
  * I tend to agree with suBN’s point 3 that it would be interesting to investigate how the proposed methods perform in higher dimensional settings. Ideally, you could present some empirical evidence that they do work well. But if this is not feasible — or if they suffer a similar curse of dimensionality — it would help to include a clearly delineated and thorough discussion of this point in the manuscript.
  * nnB2 makes some useful suggestions to increase the accessibility of the paper. I encourage you to revise the manuscript as suggested in their points 1 & 2.
  * I encourage you to investigate the sensitivity to certain tuning parameters, as recommended by nnB2’s point 5.
  * I encourage you to add a short remark about the gap between the theoretical results and practical implementation of TRUST++(tuned) as suggested by jmdt

---

> ### Author Response · Authors · 2026-03-03
>
> Thank you for the clear guidance and for summarizing the reviewers’ main points. We have revised the manuscript accordingly and respond point by point below.
>
> - Simulation cost (suBN, Point 1).
>
> We agree. Our use of data-efficient refers to requiring few real observations, not few simulations. We revised the manuscript, including the abstract, to make this explicit, and we added a short discussion noting that simulation cost can be a bottleneck when the simulator is expensive.
> - Higher-dimensional settings (suBN, Point 3).
>
> We agree this is important. We added a new experiment showing that performance degrades as dimension increases, as expected, but much more gracefully than the grid-based MC baseline. In particular, MC becomes impractical in our setup at high dimensions, while TRUST and TRUST++ remain usable with relatively small errors.
> - Accessibility (nnB2, Points 1–2).
>
> We revised the exposition to improve readability. In particular, we rewrote the TRUST/TRUST++ presentation to give more intuition for the partitioning step and the role of rho (and M), and we rewrote the nuisance-parameter section, adding a simple example to clarify the main idea.
> - Sensitivity to tuning parameters (nnB2, Point 5).
>
> We added a hyperparameter sensitivity analysis in the appendix and briefly discuss it in the main text. This shows that minimum leaf size is the most important hyperparameter, while the method is relatively robust to the others.
> - Theory vs. practice for TRUST++ (tuned) (jmdt).
>
> We added a remark clarifying that Theorem 3 applies to the strict partition variant, while the tuned TRUST++ variant may fall outside this setting when the partition assumption is not satisfied. We also edited the theorem itself.
>
> We hope these revisions address the concerns raised and improve the paper’s clarity and scope. Thank you again for the constructive feedback.

---

### Decision · Action_Editor_QLk7 · 2026-04-07

**Recommendation:** Accept as is

**Audience:**

Yes

**Audience Explanation:**

Yes: the paper considers a broadly interesting paper.

**Claims And Evidence:**

Yes

**Claims Explanation:**

The reviewers agreed that the main claims are well-supported.